 

# Using mobile phones as acoustic sensors for high-throughput mosquito surveillance

**Haripriya Mukundarajan[1], Felix Jan Hein Hol[2], Erica Araceli Castillo[1], Cooper Newby[1], Manu Prakash[2]***

[1]Department of Mechanical Engineering, Stanford University, Stanford, United States; [2]Department of Bioengineering, Stanford University, Stanford, United States

**Abstract** The direct monitoring of mosquito populations in field settings is a crucial input for shaping appropriate and timely control measures for mosquito-borne diseases. Here, we demonstrate that commercially available mobile phones are a powerful tool for acoustically mapping mosquito species distributions worldwide. We show that even low-cost mobile phones with very basic functionality are capable of sensitively acquiring acoustic data on species-specific mosquito wingbeat sounds, while simultaneously recording the time and location of the human-mosquito encounter. We survey a wide range of medically important mosquito species, to quantitatively demonstrate how acoustic recordings supported by spatio-temporal metadata enable rapid, non-invasive species identification. As proof-of-concept, we carry out field demonstrations where minimally-trained users map local mosquitoes using their personal phones. Thus, we establish a new paradigm for mosquito surveillance that takes advantage of the existing global mobile network infrastructure, to enable continuous and large-scale data acquisition in resource-constrained areas.

DOI: https://doi.org/10.7554/eLife.27854.001

## Introduction

Frequent, widespread, and high resolution surveillance of mosquitoes is essential to understanding their complex ecology and behaviour. An in-depth knowledge of human—mosquito interactions is a critical component in mitigating mosquito-borne diseases like malaria, dengue, and Zika (*Macdonald, 1956*; *World Health Organization, 2014*; *Godfray, 2013*; *Besansky, 2015*; *Kindhauser et al., 2016*). However, there is a paucity of high-resolution ecological data on the abundance, temporal variation, and spatial distribution of mosquito vector species. This poses a serious impediment to the effective control of mosquito-borne diseases (*Townson et al., 2005*; *Ferguson et al., 2010*; *Hay et al., 2013*). Efforts to map mosquito populations primarily rely on interpolative mathematical models based on factors such as clinical disease burdens or climatological data, with field inputs from entomological surveys being a comparatively sparse contribution (*Hay et al., 2010*; *Mordecai et al., 2017*). This scarcity of field data stems from the absence of high-throughput, low-cost surveillance techniques to map the distribution of mosquitoes. This effect is particularly severe in resource-constrained areas, as currently used methods like trapping and manual identification require considerable expense, labour, and time. However, for disease control strategies to be truly effective, it is critical for them to be strongly informed by current mosquito population distributions. Hence, there is a pressing need for novel methods of surveillance that can rapidly and inexpensively sample mosquito populations across large areas at high spatio-temporal resolutions.

Acoustic surveillance was proposed decades ago as a high-throughput automated surveillance approach, where mosquitoes in flight are identified using the species-specific frequencies in their wingbeat sounds (*Kahn et al., 1945*; *Kahn and Offenhauser, 1949a*; *Offenhauser and Kahn, 1949*; *Kahn and Offenhauser, 1949b*; *Jones, 1964*; *Johnson and Ritchie, 2016*). This technique is based

***For correspondence:**
manup@stanford.edu

**Competing interests:** The authors declare that no competing interests exist.

---

**eLife digest** Diseases spread by mosquito bites – like malaria, dengue and Zika – kill over half a million people each year. Many of these diseases have no cure, and those that do often face the problem of drug resistance. As such, the most effective way to control these diseases is to reduce the number of mosquitoes in the affected area. Yet, this kind of control strategy relies on knowing which species of mosquito that spread diseases to humans and where they are found.

Most current methods to monitor mosquitoes are laborious, and need expensive equipment and highly trained people. This makes them less suitable for resource-poor areas, which typically have the highest levels of mosquito-borne diseases. However, scientists have known for many decades that one can identify species of mosquito by the pitch of sound they make when they fly. The challenge was to find instruments that could record these sounds clearly enough to measure the pitch, but that were still cheap and sturdy enough to use in mosquito-infested environments.

Mukundarajan et al. now report that the microphones in ordinary mobile phones are sensitive enough to accurately record the whining buzz of a mosquito, even in the presence of background noise. First, techniques were developed that allow anybody to record mosquitoes on virtually any mobile phone model – including smartphones and basic flip-phones. Next, the sounds of 20 species of mosquito that spread diseases to humans were recorded – more than in any previous study. When other recordings made using various phones were compared to this collection of sounds, the correct species was automatically identified for two thirds of the recordings. This meant that, together with volunteers who were trained for 15 minutes, Mukundarajan et al. could use mobile phone recordings to map mosquito species inside a National Park in California and a village in Madagascar more efficiently than with any conventional method.

There are over five billion mobile phones in use today. Even a small percentage of mobile users recording mosquito sounds could thus provide a tremendous amount of data. As more and more people contribute recordings, Mukundarajan et al. hope to be able to make real-time global maps of mosquito distributions. These could then be used to design better strategies to control mosquitoes and combat mosquito-borne diseases.

DOI: https://doi.org/10.7554/eLife.27854.002

---

on the hypothesis that sexual selection has led to characteristic sound signatures for different mosquito species (*Roth, 1948*; *Belton, 1994*; *Gibson and Russell, 2006*; *Cator et al., 2009*; *Warren et al., 2009*; *Pennetier et al., 2010*; *Cator et al., 2010*). However, the challenge of using expensive microphones to acquire low amplitude mosquito sounds against potentially high background noise poses a significant barrier to its widespread adoption as a field technique (*Jones, 1964*; *Unwin and Ellington, 1979*; *Raman et al., 2007*). Optical wingbeat measurement has been proposed as a proxy for sound recording, to overcome the technological limitations in audio signal acquisition (*Moore et al., 1986*; *Batista et al., 2011*; *Chen et al., 2014*; *Silva et al., 2015*). However, this approach is yet to take flight on a large scale, as the cost and global scalability of the requisite hardware remain challenges for its worldwide deployment.

Here, we propose a novel solution that uses mobile phones to enable widespread acoustic mosquito surveillance. We use the sensitive microphones in mobile handsets to record species-specific wingbeat sounds from a variety of mosquito vector species for automated identification and analysis (*Figure 1A*). Our technique exploits the rapid advances in mobile phone hardware technology, recognizing that these highly portable devices are optimized for sophisticated audio capture, processing, and transmission. Further, the proliferation of the global mobile network implies that around 5.3 billion users worldwide are connected today by a distributed data transmission infrastructure, growing quickly at over 6% annually (*Cerwall et al., 2017*). Mobile connectivity has revolutionized the delivery of services based on mapping and analyzing crowdsourced data from mobile phones, which has enabled many innovative applications in citizen science (*Graham et al., 2011*; *Catlin-Groves, 2012*; *Malykhina, 2013*; *Kong et al., 2016*). Also, the explosive growth in mobile phone use is most pronounced in Africa, Asia, and Latin America (*Cerwall et al., 2017*), which also bear the brunt of mosquito-borne diseases (*World Health Organization, 2014*). This overlap implies that a mobile phone based concept can be particularly scalable, sustainable, and cost effective for

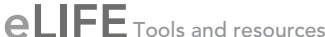

**Figure 1.** Mobile phone users can collect acoustic data from mosquitoes characterized by the base frequency and harmonics. (**A**) Illustration showing the collection of mosquito acoustic data by mobile phone users in different locations. (**B**) Techniques to acquire wingbeat sounds from mosquitoes using mobile phones include lab methods like (i) collecting them in cages, and field methods like (ii) following mosquitoes in free-flight, or (iii) capturing them in cups, bottles or inflated bags. (**C**) Spectrogram for a flight trace acquired from an individual female *Anopheles gambiae* mosquito using a 2006 model Samsung SGH T-209 flip phone. The wingbeat frequency at every instant is computationally identified and marked with a black line. (Top) The time-averaged spectrum of this flight trace shows the distribution of acoustic power among the base frequency and multiple harmonics. (**D**) The variations in wingbeat frequency of the mosquito during this flight trace are represented by a probability distribution of the frequency identified in each window of the spectrogram. (Top) Raw wingbeat frequency data are represented as a violin plot with an overlaid box plot marking the inter-quartile range, black circle representing mean frequency, gray vertical bar for median frequency, and whiskers indicating 5th and 95th quantiles.

DOI: https://doi.org/10.7554/eLife.27854.003

The following figure supplements are available for figure 1:

**Figure supplement 1.** Schematic of proposed surveillance system using crowdsourced acoustic data from mobile phones.

DOI: https://doi.org/10.7554/eLife.27854.004

**Figure supplement 2.** Brochure.

DOI: https://doi.org/10.7554/eLife.27854.005

sampling real-time mosquito population dynamics in resource-constrained areas with high disease burdens. Our proof-of-concept study highlights the potential of our solution to engage citizen scientists around the world in mosquito surveillance, without the need for specialized equipment or extensive formal training.

## Results

### Mobile phone microphones can be used to record wingbeat sounds from mosquitoes

We record the acoustic signatures of free-flying mosquitoes using mobile phones, by orienting the primary microphone towards the mosquito and using an audio recording application to acquire and store its wingbeat sound (*Figure 1A,B*, *Figure 1—figure supplements 1* and *2*, *Supplementary file 1*, *Videos 1–3*). Mosquito sounds have relatively low complexity. They comprise a single fundamental frequency with several overtones, which we identify using the short-time Fourier transform (STFT) (*Figure 1C*, *Video 4*). The sound from typical flight traces is not monotone, but shows natural frequency variations which we capture as a wingbeat frequency distribution, in a range characteristic of the given species (*Figure 1D*). Since mosquitoes rarely fly at speeds over half a meter per second, the Doppler shift of frequency during free-flight $(1 - [330 - 0.5/330 + 0.5] \approx 0.3\%$, i.e. <2 Hz) is small when compared to the observed natural spreads of up to 100 Hz within a single flight trace. Sexual dimorphism in most species implies that males have higher frequencies than females. Female wingbeat frequencies are typically between 200 and 700 Hz. This overlaps the voice frequency band (300 to 3000 Hz), in which phones are designed to have maximum sensitivity. In addition, using mobile phones as recording platforms automatically registers relevant metadata parameters, such as the location and time of data acquisition. This adds valuable secondary information to acoustic data that is useful for species identification and spatio-temporal mapping. Such acoustic and spatio-temporal information could potentially be aggregated from many mobile phone users, to generate large data sets that map the distribution of mosquito species at high resolutions (*Figure 1—figure supplement 1*).

### Mobile phone microphones accurately capture wingbeat frequencies without distortion

Our concept of mobile phone based acoustic surveillance is based on the idea that mobile phones are high fidelity acoustic sensors, which faithfully capture the frequencies produced by mosquito wingbeats during flight. To confirm this, we compared measurements of wingbeat frequency from two independent modalities, high speed video and mobile phone audio (Sony Xperia Z3 Compact), for female *Culex tarsalis* mosquitoes in tethered flight (*Figure 2A*, *Video 4*, *Video 5*). For synchronized audio-video recordings, spectrograms were aligned in time to within 2 ms, and show an exact match in frequency at each time window to within a computational error margin of 2 Hz (*Figure 2B*). The

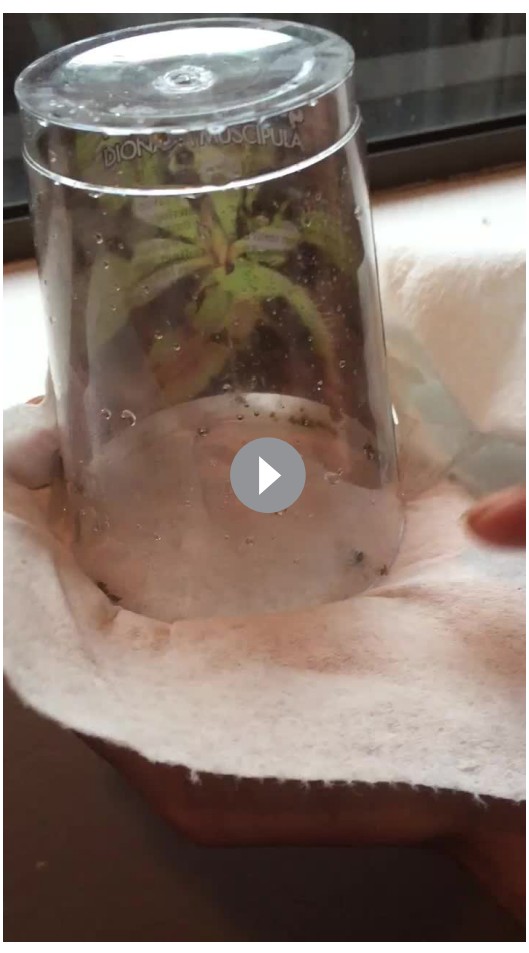

**Video 1.** Demonstration of how to identify the primary microphone of a mobile phone and record sound from a mosquito confined in a cup.
DOI: https://doi.org/10.7554/eLife.27854.006

respective distributions of wingbeat frequency have maximum sample fractions occurring in the same frequency bin (*Figure 2C*), giving the same median frequency of 264 Hz, despite the audio signal being noisier than the video. To show similarity between distributions, we compute the Bhattacharya Coefficient (BC) - a measure of the mutual overlap of two distributions, ranging between 1 for completely coincident and 0 for completely separated distributions. We obtain a high BC of 0.95, implying an almost complete overlap between the distributions. This corroborates the spectral accuracy of wingbeat frequency distributions recorded by mobile phones, based on an independent optical reference.

## Mobile phone microphones are comparable to studio microphones in recording mosquito wingbeats at up to 100 mm

We propose using mobile phones for acoustic surveillance under field conditions, where the primary challenge is recording faint mosquito sounds in noise-prone environments. Accordingly, it is important to establish the limits within which their built-in microphones are sensitive enough to reliably acquire data with high signal-to-noise ratios (SNR). The technical specifications of many commercial mobile phone microphones are not openly available. Hence, we experimentally calibrated a range of mobile phone models against two reference electret condenser microphones under identical conditions. Using a piezoelectric buzzer of constant amplitude (77 dB at source) and frequency (500 Hz) as a standardized sound source, we first showed that both smartphones (iPhone 4S, Xperia Z3 Compact) and low-end feature phones (Samsung SGH T-209 clamshell model) have SNRs comparable to the reference microphones over distances up to 100 mm (*Figure 2D*). Next, we recorded wingbeat sound from tethered mosquitoes to gauge suitable working distances over which SNR is high enough to allow detection. Curves of mobile phone SNR over distance indicate that all the phones tested, including a decade-old basic phone (SGH T-209), are capable of acquiring detectable wingbeat sound up to 50 mm from a mosquito (*Figure 2E*). This is a working distance we have found to be easily achievable when making free-flight recordings in the field. This distance improves to 100 mm in the case of smartphones like the Xperia, making it even easier for users to record mosquito sound (*Figure 2—figure supplement 1*). Beyond these distances, the mosquito sound becomes imperceptible, drowned out by background noise. We also caution that these working limits are appropriate in quiet to moderately noisy environments with background noise levels below 50 dB, and shrink drastically in louder conditions. In addition to distance, the orientation of the mobile phone is also critical to maximizing SNR, as its acoustic sensitivity is directional and localized around the primary microphone. Thus, the success of using mobile phones to record mosquitoes is fundamentally dependent on active involvement from the user, to maintain the appropriate distance and orientation of the phone with respect to the mosquito, and avoid introducing noise via bumps and scrapes of the microphone.

## Diverse mobile phone models acquire quantitatively equivalent acoustic data from mosquitoes

There are over 100 mobile phone brands in use worldwide, with a range of devices widely varying in features and capability. Widespread mobile phone-based surveillance is only possible if mosquito wingbeat sound collected using any phone can be analyzed using the same identification metrics and algorithms. This means that a diverse range of phone models must record quantitatively equivalent acoustic data with identical spectral profiles. This is a necessity for our proposed surveillance technique to scale to the broadest possible user base, particularly in countries with lower smartphone penetration. We compared eight different smart and feature phones ranging in price between ~$20 and ~$700 (*Figure 2F*), by analyzing 5 minutes of audio recorded with each phone from free-flying female *Anopheles stephensi* mosquitoes of similar age from a lab-reared population contained in a cage. We noted high degrees of overlap in wingbeat frequency distributions recorded by each phone for this population, with both mean and median frequencies obtained by each phone differing by less than 5% from other phones (*Figure 2F*).

We computed the Bhattacharya Coefficient (BC) and obtained high degrees of overlap between 0.93 and 1 (*Figure 2H*). We also quantified how distinguishable these wingbeat frequency distributions are, using the Jensen-Shannon divergence metric (JSD) between each pair of phones (*Figure 2G*). This metric, which ranges from 0 for identical to 1 for completely disparate

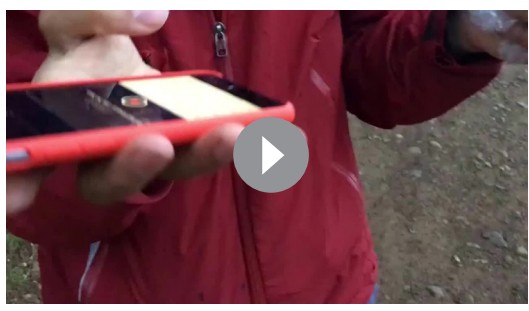

**Video 2.** Demonstration of how to record sound from a mosquito confined in a bottle, using a mobile phone.
DOI: https://doi.org/10.7554/eLife.27854.007

distributions, can be inversely related to the amount of mutual information shared between the distributions. The JSD for each pair of phones had low values below 0.3 indicating high similarity in measurement, corroborating that wingbeat frequency sampling is relatively insensitive to the phone used. This is also true of other species we surveyed, with the JSD between phones typically between 0.1 and 0.35 for a given population. This range of JSD provides an estimate of the variance inherent in sampling a population of the same species. Hence, the upper end of this range gives an approximate quantitative threshold for differentiability requirements between wingbeat frequency distributions of different species. Thus, we demonstrate that a diverse range of both smart and feature phones provide highly similar acoustic spectra from the same population of mosquitoes, enabling a truly universal acoustics-based platform for mosquito identification.

## Mosquito species have characteristic wingbeat frequency distributions that are measurable using mobile phone recordings

The most important ingredient for acoustic identification of mosquitoes is prior knowledge of the wingbeat frequency distributions corresponding to different species. We surveyed representative populations from 20 medically relevant mosquito species under similar experimental conditions (*Figure 3A*), to establish a reference dataset for classifying acoustic recordings (*Mukundarajan et al., 2017*). Although we recorded mosquitoes using a variety of phones, the results presented here focus on data acquired by the 2006-model $20 SGH T-209 feature phone. This highlights that even a basic low-end phone is capable of collecting high-quality acoustic recordings, to precisely measure wingbeat frequency distributions.

We analyzed the recordings of the reference populations by dividing their spectrograms into 20 ms sample windows, and applying a peak-finding algorithm in each window to measure instantaneous frequency. The number of frequency values we obtained for each species had a wide range between 300 and 50,000 data points, and these were binned into wingbeat frequency distributions (*Figure 3A*). To verify whether these reference distributions are sufficiently well-sampled, we drew random subsets of frequency values from each distribution, and measured the statistical distance from these subsets to their parent distributions. For subsets of increasing size, we observed a rapid convergence of the BC and JSD to their respective values of 1 and 0 expected for distributions identical to the parent. We showed computationally that the smallest subset of randomly selected values which has a BC above 0.9 as compared to the

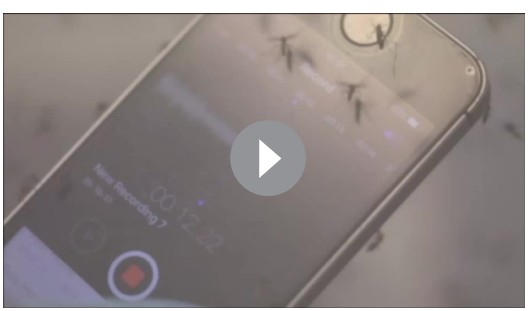

**Video 3.** Demonstration of how to record sound from a mosquito in free flight, at the moment of take-off, using a mobile phone.
DOI: https://doi.org/10.7554/eLife.27854.008

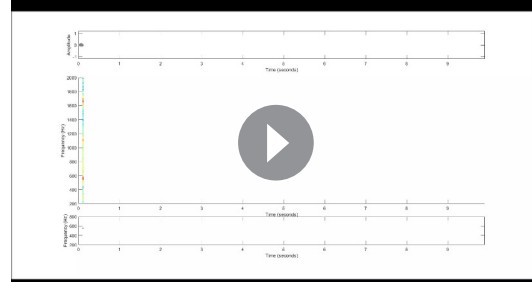

**Video 4.** Video showing variation of amplitude with time, spectrogram, and automatically identified wingbeat frequency at eachtime window, synchronized with the Anopheles gambiae female mosquito recording corresponding to *Supplementary file 1*.
DOI: https://doi.org/10.7554/eLife.27854.011

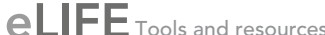

**Figure 2.** Mobile phones sensitively acquire high fidelity acoustic data from mosquitoes with comparable performance across models. (**A**) Schematic of experimental setup for recording a tethered mosquito using synchronized mobile phone audio and high speed video. Synchronization between audio

*Figure 2 continued on next page*

*Figure 2 continued*

and video is achieved on the order of microseconds, using a piezoelectric buzzer connected in parallel with an LED, and controlled with a microprocessor to produce the same temporal sequence of sound and light. (B) Overlaid spectrograms for female *Culex tarsalis* mosquitoes obtained independently using high speed video (magenta) and mobile phone audio (cyan), aligned to within 2 ms and showing a spectral overlap (blue) within 2 Hz across all time instances. The mobile phone data is noisy but faithfully reproduces the wingbeat frequency peak of 264 Hz and the first two overtones. (C), Wingbeat frequency distributions from video and audio (165 time instances each) have high overlaps, as measured by a Bhattacharya Coefficient (BC) of 0.95. (D) Signal-to-noise ratio (SNR) estimates over distance from a standardized sound source show that mobile phone microphone performance within a 100 mm radius is superior or comparable to high performance studio microphones. The pink line represents the actual source amplitude, with the pink-shaded region below indicating the region less than the actual amplitude, in which all acoustic measurements should lie. The gray-shaded band represents the range of sensitivities between the limits set by the two studio microphones used as reference standards. The SNR curves for all three phones lie mostly within this band, indicating that they perform in a range between the two reference microphones. (E) SNR over distance for the wingbeat sound produced by a tethered female *Cx. tarsalis* mosquito (normalized for a source amplitude of 45 dB), providing working limits where phones can detect the audio signal - 50 mm for the low end T-209 feature phone and 100 mm for the iPhone 4S and Xperia Z3 Compact smartphones. The gray-dotted line represents the actual amplitude of the mosquito sound in dB, as measured by the MXL991 reference microphone. The gray-shaded region below indicates the region less than the actual amplitude, in which all acoustic measurements are expected to lie. (F) Variation of the wingbeat frequency distribution sampled by 8 different phones is low compared to the natural variation within a population of lab-reared *Anopheles stephensi* females. Raw data are shown with overlaid box plots marking the inter-quartile range, black circles for mean frequency, gray vertical bars for median frequency, and whiskers indicating 5th and 95th quantiles. (G,H) The Jensen-Shannon divergence metric for base frequency distributions (G, lower left triangle) shows low disparity, ranging between 0.144 and 0.3, against a minimum of 0 for identical distributions. Likewise, the Bhattacharya distance (H, upper right triangle) shows high overlap, with values between 0.935 and 0.986, against a maximum of 1 for identical distributions. The brown-hatched areas along the diagonal represent blank cells, as distances are not shown for any distribution with respect to itself.
DOI: https://doi.org/10.7554/eLife.27854.009

The following figure supplement is available for figure 2:

**Figure supplement 1.** Synchronized recordings of tethered mosquitoes using studio and mobile phone microphones show exact correspondence at distances below 50 mm.
DOI: https://doi.org/10.7554/eLife.27854.010

full distribution still has a length which is a small fraction (less than 0.2) of the total number of samples in the full distribution for each species. Some species, such as *An. dirus* and *An. stephensi*, show slower convergence, due to the smaller sample sizes used to build the frequency distributions. The time for convergence above a BC of 0.9 corresponds to approximately 60 samples, or 1.2 seconds of recording time, to generate a well-sampled wingbeat frequency distribution for a given population. Since our reference distributions were recorded for between 6 and 1000 seconds each, we conclude that they are sufficiently representative of the variations within the reference population. Nevertheless, it is also clear that the longer a population is sampled for, the better the frequency distribution constructed to represent it.

We also considered whether significant variations exist between different colonies or populations within a species. We found that different lab-reared strains of *An. gambiae*, including permethrin susceptible and resistant variants sourced from Kenya and an additional bendiocarb resistant strain sourced from Benin, produced highly similar frequency distributions with large overlaps of 0.84 to 0.89 as measured by their mutual BCs (*Figure 3—figure supplement 1C*). Similarly for *Aedes aegypti*, lab-reared mosquitoes from two separate colonies of the same New Orleans origin, as well as first generation progeny from eggs retrieved from the field in Los Angeles, had high overlaps in their frequency distributions with mutual BCs ranging between 0.66 and 0.74 (the slightly lower BC values arise due to the distribution for one of the colonies having a smaller spread despite being completely overlapped by the other two) (*Figure 3—figure supplement 1C*). Thus, we hypothesize that it may be possible for certain species to have wingbeat frequency distributions that are universal across colonies and strains, facilitating automated identification using the same reference. However, we also found that lab-reared strains of *An. arabiensis* originally sourced from Senegal and Sudan have non-overlapping frequency distributions, with a very small BC of 0.14 (*Figure 3—figure supplement 1C*). This signals that there may be other factors underlying geographic variability in wingbeat frequency within a species, which merit further investigation. Hence, it is advisable to build local acoustic databases wherever possible in order to improve classification, by recording actual frequency distributions from mosquitoes whose species have been verified by non-acoustic methods (*Figure 1—figure supplement 1A–D*).

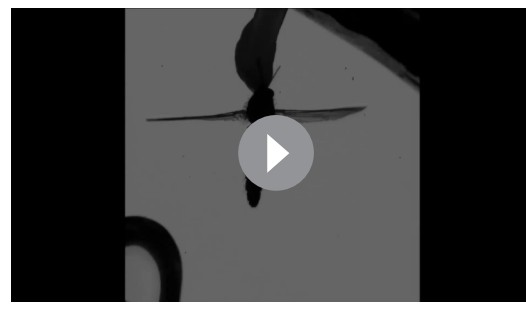

**Video 5.** High-speed video recording from a lab reared female *Culex tarsalis* mosquito, tethered by the scutum to a pipette tip using a bead of wax. The calibration LED visible at the bottom left is used to automatically synchronize time points in the video with corresponding beeps from a piezoelectric buzzer in the audio. Recording was done on a Phantom v1610 high-speed camera sampling at 10 kHz, with playback slowed down 100 times.

DOI: https://doi.org/10.7554/eLife.27854.012

## Estimation methods can be used for species classification of acoustic recordings

The core concept of acoustic surveillance is the correct identification of the species of a mosquito given its wingbeat acoustic signature. Although previous approaches have used a single averaged value of frequency to represent a flight trace (*Moore et al., 1986*; *Batista et al., 2011*; *Chen et al., 2014*; *Silva et al., 2015*; *Villarreal et al., 2017*), our approach takes into account the natural variance within each trace, by treating the collection of frequency values across all time windows of the recording as a distribution. We classify a given flight trace by statistically comparing this distribution against the known wingbeat frequency distributions for different species. Although classification is possible through several approaches that emphasize different statistical comparisons, in this work we apply the method of Maximum Likelihood Estimation (MLE). Here, the species is considered to be a parameter, whose discrete values affect the statistical distribution of wingbeat frequency over a specified frequency range. We classify a given flight trace as belonging to that species whose corresponding wingbeat frequency distribution is most likely to have produced the observed frequency values from that trace, with a confidence level equal to the likelihood estimate. The reference distributions for comparison are built using the data presented in *Figure 3A*, collected using the SGH T-209 basic phone.

The best indicator for how likely it is for recordings from a given species to be correctly classified is the degree of overlap of its characteristic wingbeat frequency distribution with other potentially confounding species. One way to measure this is the mutual JSD between pairs of species, which must be significantly higher than the threshold of 0.35 established earlier for the variance inherent within the same species. The vast majority of all possible pairwise combinations of species in our study (184 out of 190) had JSD greater than this minimum value (*Figure 3B*), indicating that differences in frequency distribution between species are significant in most cases (*Figure 3C*). We translate this into probabilistic estimates of classification accuracy, based on the principle that randomly drawn samples bootstrapped from a given species' frequency distribution will be classified with an error rate that reflects its overlap with other species' distributions. We applied the MLE algorithm on bootstrapped subsets from the wingbeat frequency data used to create the reference distributions, to build a confusion matrix showing the relative proportions in which subsets derived from different species are classified correctly or erroneously. Our results verify that species with fairly unique and non-overlapping wingbeat frequency distributions are almost always correctly classified, while those with high degrees of overlap are most likely to be mutually misclassified. This provides quantitative estimates of the inherent degree of differentiability of each species with respect to all others (*Figure 3D*), setting intrinsic limits on the success of any classification scheme based exclusively on wingbeat frequency.

The classification accuracies and error rates for samples bootstrapped from the same data used to build reference distributions represent a best case scenario. A more realistic test of our classification approach is its ability to handle new flight traces that are not part of the original reference datasets. We validated the MLE algorithm by classifying recordings from the iPhone 4S, Sony Xperia Z3 Compact, and Google Nexus One (*Figure 3E*, *Figure 3—figure supplement 1D*), by comparing them to the reference dataset collected using the SGH T-209 (*Figure 3A*). This is a validation dataset that has similar species frequency distributions as the reference dataset, but also incorporates the additional variance arising from using different phones. We sequentially tested a large number of individual flight traces chosen at random from eight different species, to simulate the stochastic

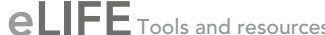

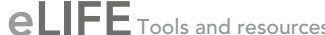

**Figure 3.** Mosquitoes of different species are distinguishable based on wingbeat frequency distributions and metadata. (**A**) Distribution of wingbeat frequencies for female mosquitoes of 20 vector species, for recordings obtained with the 2006 model SGH T-209 low-end feature phone (except *Culex*

*Figure 3 continued on next page*

*Figure 3 continued*

*pipiens, Culex quinquefasciatus*, and *Culiseta incidens,* recorded using iPhone models; and *Aedes sierrensis*, recorded with various phones). (**B**) (lower left triangle), Jensen-Shannon divergence metric for base frequency distributions. Distributions are spaced apart with high J-S divergence in most cases, with only four pairwise combinations having J-S divergence around 0.35 - the maximum divergence for the same species across different phones. (**C**) (upper right triangle), Qualitative classification of species pairs according to the possibility of distinguishing them using mobile phones — (i) no frequency overlaps, hence distinguishable by acoustics alone, (ii) overlapping frequency distributions, but not geographically co-occurring hence distinguishable using location, (iii)overlapping frequency distributions but distinguishable using time stamps, (iv) partially overlapping frequency distributions but no location-time distinctions, hence distinguishable but not in all cases, (v) indistingishable due to highly overlapping frequency distributions with co-occurrence in space and time. The brown-hatched areas along the diagonal represent blank cells, as distances are not shown for any distribution with respect to itself. (**D**) Confusion matrix for classification of acoustic data bootstrapped from the reference frequency datasets for twenty species, using the Maximum Likelihood Estimation (MLE) algorithm. Each column corresponds to a particular species, from which acoustic data is drawn for classification. The twenty entries in each column represent the fractions in which recordings from the species corresponding to that column are classified among all twenty species in our database. This classification is done for data taken from the reference distributions, and is compared against the same reference dataset, based exclusively on wingbeat frequency. Classification errors occur when a given species frequency distribution has overlaps with other species, and the confusion matrix reflects the inherent uniqueness or overlap between frequency distributions in our database. Colour scale showing fraction of recordings classified is the same for both D and E. (**E**) Confusion matrix for classification of test audio recordings, using the MLE algorithm. Test data were collected for 17 species, using different phones (some or all among Google Nexus One, Sony Xperia Z3 Compact, iPhone 4S) which were not used to construct the reference distributions (recorded using the SGH T-209). Each column corresponds to a particular test species, from which acoustic data are drawn for classification. The twenty entries in each column represent the fractions in which recordings from the test species corresponding to that column are classified among all twenty species in our database. Classification is based on both wingbeat frequency and a location filter, simulating the classification of randomly recorded mosquitoes of different species by users in field conditions. The resulting classification accuracies are significantly higher in this case, when compared to classificadion accuracies which do not consider location (*Figure 3— figure supplement 1D*). Blank areas along three columns represent species for which test data from a different phone were not available. (**F,G**) Variations in base-frequency distribution (F) for field-recorded sounds corresponding to wild female *Ae. sierrensis* mosquitoes having a wide (about two-fold) variation in body size and wing area (G), showing small differences between individuals compared to the variation within each flight trace. The gray distribution at the top represents the species wingbeat frequency distribution for *Ae. sierrensis*, with the gray-shaded vertical band marking the range from 5th to 95th percentile of frequency for the species. All individual recordings lie completely within this range.

DOI: https://doi.org/10.7554/eLife.27854.014

The following figure supplements are available for figure 3:

**Figure supplement 1.** Statistical parameters for adequate sampling, inter-colony variations and test data classification accuracies for wingbeat frequency distributions.
DOI: https://doi.org/10.7554/eLife.27854.015

**Figure supplement 2.** Mosquito species can be distinguished with mobile phone acoustics and metadata.
DOI: https://doi.org/10.7554/eLife.27854.016

nature of human-mosquito encounters involving a variety of species in the field. Each flight trace was compared against all species in the reference dataset, and the best match was chosen. We observe that classification accuracies are slightly lower than those obtained for bootstrapped reference data (*Figure 3E*, *Figure 3—figure supplement 1D*), indicating that our MLE approach is inherently limited in practice by overlaps in wingbeat frequency distributions.

## Acoustic data can be combined with metadata to improve classification accuracies

The limitations of using wingbeat frequency as the sole distinguishing characteristic between species are exposed by the inability to perfectly classify species that have overlapping frequency distributions. A typical wingbeat frequency distribution for any given species has a spread of 150 to 200 Hz between the 5th and 95th percentiles, located within the frequency range of 200 to 700 Hz for most female mosquitoes. But with over 40 species of medical relevance among *Anopheles* mosquitoes alone, the pigeon-hole principle makes it impossible to have perfectly non-overlapping distributions for all species (*Chen et al., 2014*). Unlike speech (*Rabiner and Schafer, 2011*), music (*Wang, 2003*), or other insects (*Drosopoulos and Claridge, 2005*), birds (*Marler and Slabbekoorn, 2004*) and animals (*Mellinger et al., 2007*; *Mellinger and Clark, 2000*; *Portfors, 2007*), the low time-domain complexity of mosquito sound makes it difficult to identify temporal motifs or other acoustic parameters that can be used in species identification (*Chen et al., 2014*; *Kiskin et al., 2017*). To improve the classification of mosquito species, *Chen et al. (2014)* introduced the concept of metadata, where parameters like the time or place of recording can be used as additional features to differentiate between mosquitoes with varying circadian activity or geographic distribution. Mobile phones

offer the advantage of automatically registering time stamps for acoustic data, and location tags from cell tower or GPS measurements, along with sensors to collect other metadata such as photographs which can support identification. We assessed the effect of including even a single metadata parameter to refine species classification, by applying a location filter based on entomological survey data from over 200 countries for the 20 species in our database (*Walter Reed Biosystematics Unit, 2017*). These data are available in the public domain (*Walter Reed Biosystematics Unit, 2017*), and can be readily incorporated as prior knowledge to aid in classification. We repeated our sequential classification of flight traces from randomly chosen species, with each recording randomly assigned to one of the countries where that species is known to be found. Recordings are then compared only among the subset of species co-located in that country. This significantly improved classification accuracy not only for individual species, but also from about 35% to 65%, as an average over all the species tested (*Figure 3E*, *Figure 3—figure supplement 1D*). In earlier work, algorithms for mosquito species classification have achieved accuracies of around $80\%$, when comparing a smaller number (10) of insect types. This included both sexes of four acoustically dimorphic mosquito species, and two species of flies (*Chen et al., 2014*). Although the overall classification accuracy by our method is a little lower at 65%, we note that this is for twice the number of species and includes only female mosquitoes. Since classification accuracy decreases significantly with the number of species considered and their degree of frequency overlap, the accuracies produced by our classification algorithm appear to be reasonable in comparison to earlier work. We underline the caveat that in practice, classification accuracies can be further lowered by species other than the ones in our reference database. However, the inclusion of a location filter eliminates potential errors from irrelevant species, and highlights the discriminatory power of such metadata.

For greater clarity, we illustrate a few scenarios where acoustic data can be combined with metadata for identifying mosquitoes in field investigations (*Figure 3—figure supplement 2*). In the simplest case, species with completely non-overlapping frequency distributions, such as *Anopheles gambiae* and *Culex pipiens* (JSD = 1, negligible probability of mutual mis-classification), can easily be distinguished by sound alone (*Figure 3—figure supplement 2A*). Some species with overlapping frequency distributions can still be perfectly classified using location metadata on the basis of their spatial segregation. For instance, *An. atroparvus* has a 38% probability of being mis-classified as *An. dirus* (overlapping frequency distribution with mutual JSD $= 0.26 < 0.35$). However, this confusion never occurs when location is taken into account, as these species are respectively found in Europe and South Asia (*Sinka et al., 2012*), and do not co-exist in the same area (*Figure 3—figure supplement 2B*). The use of timestamps has been suggested previously in the case of species which are overlapping in both frequency and spatial distribution, but have different circadian activity (*Chen et al., 2014*; *Silva et al., 2015*). As an example, *Ae. aegypti* is often misclassified as *An. gambiae* with high probability based on wingbeat frequency (mutual JSD $= 0.37 \sim 0.35$, *Ae. aegypti* wrongly classified 95% of the time). However, classification accuracy may possibly be substantially improved if recordings collected during daytime were all classified as *Ae. aegypti*, since it is much more likely to be active during daylight hours (*Figure 3—figure supplement 2C*). Co-occurring and morphologically similar species of interest that can be acoustically distinguished include the arboviral vectors *Ae. aegypti* and *Ae. albopictus* (*Brogdon, 1994*) (JSD $= 0.55 > 0.35$, correctly distinguished by MLE around 98% of the time), and the closely related species *Cx. pipiens* and *Cx. quinquefasciatus* (JSD $= 0.65 > 0.35$, correctly distinguished around 95% of the time) (*Figure 3—figure supplement 2D*). Morphologically indistinguishable vector species like members of the *Anopheles gambiae s.l.* complex are of particular interest for acoustic identification (*Brogdon, 1998*; *Wekesa et al., 1998*; *Tripet et al., 2004*). Our results for four members of this complex imply significant distinguishability for *An. arabiensis*, *An. quadriannulatus*, *An. gambiae*, and *An. merus*. The JSD for all but one species pair ranges between 0.61 and 0.91 (*Figure 3—figure supplement 2E*), and *An. gambiae* and *An. quadriannulatus* are correctly identified with over 97% probability. However, the pair of *An. arabiensis* and *An. merus* has high frequency overlap (JSD $= 0.29 < 0.35$, *An. arabiensis* is misclassified as *An. merus* with around 15% probability) (*Figure 3—figure supplement 2F*). Improving classification rates in this case may not be possible without additional knowledge of their specific local ecology, such as the distribution of saltwater breeding sites for the halophilic *An. merus*. Despite such limiting cases, the wingbeat frequency combined with phone provided metadata has significant discriminatory power between species in many situations, making mobile phone-based acoustic surveillance an extremely promising screening tool for mosquito population compositions (*Figure 3C*).

## Mobile phones can be effectively used to map mosquito activity through user-driven acoustic surveillance

So far, we have calibrated the performance of mobile phones and studied acoustic frequency dispersion among different mosquito species, using lab-reared populations in controlled environments. We now test our mobile phone based surveillance approach under field conditions, by collecting mosquito wingbeat data in various settings and measuring acoustic variance among wild mosquito populations. Using the techniques described earlier (*Figure 1—figure supplement 2*, *Videos 1–3*), we recorded mosquitoes not only in free flight, but also under confinement in cups or bottles to amplify the sound and increase the length of flight traces, due to the enforced proximity to the microphone. We also collected recordings in a wide range of ambient environments, from urban and rural homes, both indoors and outdoors, to forests and parks (*Figure 4—figure supplement 1*, *Supplementary files 2–7*). We obtained mosquito recordings with high SNR, whose distinctive narrow-spectrum characteristics allow their easy visual identification in spectrograms against background noises such as birdsong, fire truck sirens, and human speech. We matched the sound signatures against our frequency distribution database (*Figure 3A* for females, data for males not shown) to identify their species, which we also confirmed by capturing the respective specimens for morphological identification by microscopy.

Wild mosquitoes in field environments may vary considerably in terms of age, body size, rearing temperature, and nutrition status, yet our method requires the variation in their wingbeat frequencies to be sufficiently small in order to be identifiable. We explored the variations in wingbeat frequency for flight traces corresponding to over 80 individual free-flying *Aedes sierrensis* mosquitoes, recorded by volunteers at Big Basin Redwoods State Park in California, USA (BBR) (*Figure 4—figure supplement 2A*). Wingbeat frequency distributions for a population typically span a characteristic frequency range with a spread of around 150 to 200 Hz (*Figure 3A*). For individual flight traces collected in the field, the distributions lie at different positions within the population-level frequency range, and have spreads (difference between 5th and 95th percentile) between 10 and 100 Hz (*Figure 4—figure supplement 2B,C*). The spread in frequency for a single flight trace is typically smaller than the spread across a population, with the difference between mean frequencies for different traces contributing to the additional variation in the population's frequency distribution. This spread in each flight trace also appears uncorrelated to the duration of the trace itself (*Figure 4—figure supplement 2B,C*), suggesting that it may arise due to aerial manoeuvers during flight (*Video 6*). Although the mean frequency for most individuals is tightly clustered, there are still outliers that pose challenges for classification (*Figure 4—figure supplement 2B*). Additionally, the classification accuracy for any given recording was not a function of the duration of the trace but of the mean frequency, indicating that overlaps between frequency distribution play the fundamental role in identifying the species correctly, no matter how long the flight trace is recorded for (*Figure 4—figure supplement 2D*). We also observed that frequency distributions for individual flight traces have a minimum spread of around 2% of the mean frequency irrespective of the mean frequency or the duration of the recording, suggesting that this is an inherent natural variation in the wingbeat frequency during free flight (*Figure 4—figure supplement 2C*). Further, both the spread and the duration of the recording did not affect whether the recording was classified correctly (*Figure 4—figure supplement 2E*), corroborating the importance of wingbeat frequency as the primary identifying factor.

Wingbeat frequency for insects, including mosquitoes, has been shown to vary linearly with temperature (*Oertli, 1989*; *Villarreal et al., 2017*). It may be possible to correct for this factor by measuring ambient temperature when recording as part of relevant metadata, as suggested in other studies (*Chen et al., 2014*). We

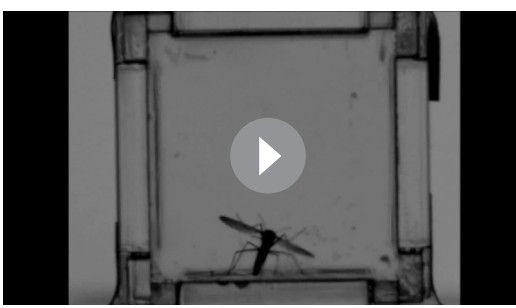

**Video 6.** High speed video recording from a lab-reared Culex tarsalis mosquito in free-flight, contained within a clear acrylic cage. Recording was done on a Phantom VEO high-speed camera sampling at 10 kHz, with playback slowed down 100 times.
DOI: https://doi.org/10.7554/eLife.27854.013

also investigated correlations between wingbeat frequency and body size, by comparing the physical specimens corresponding to several flight traces (*Figure 3F,G*). For mosquitoes varying almost two-fold in size (*Figure 3G*), the difference in mean frequency between each mosquito is less than the spread within each individual flight trace, and non-monotonic with body size (*Figure 3F*). This provides a field confirmation of lab studies that show no relationship between wingbeat frequency and body size (*Villarreal et al., 2017*). Despite the variations among individuals, our MLE approach (using a pool of all field recordings together to create a reference frequency distribution) was able to correctly classify recordings from wild *Aedes sierrensis* mosquitoes in about two-thirds of cases (54 out of 82 recordings), when considering other local species (*Cx. pipiens*, *Cx. quinquefasciatus*, *Cx. tarsalis*, and *Cu. incidens*) as potential confounders.

We next show how mobile phone recordings of mosquitoes in the field can be pooled from minimally trained users for spatio-temporal mapping, using the time and location metadata associated with each recording. Our recordings of wild *Ae. sierrensis* mosquitoes in BBR (*Figure 4—figure supplement 2A*) represent the only species found in the park, based on microscopic identification of the corresponding captured specimens. These were collected over a continuous 3 hour period by 15 volunteers hiking along trails and recording with their personal mobile phones. The recordings were mapped according to their locations to reflect spatial variations in mosquito distribution within the park (*Figure 4D*). Parsing this data by recording time further reveals the rise and fall in the number of human-mosquito encounters over an evening (*Figure 4D* inset), which can be correlated to the circadian activity of *Ae. sierrensis*.

In addition to the temperate forest environment represented by BBR, we carried out another small-scale field trial in a tropical village, at Ranomafana (District Ifanadiana) in Madagascar (RNM). Here, our first step was to establish the baseline species composition, by collecting live mosquitoes in traps or bags and identifying them using microscopy. We found both *Anopheles* and *Culex* mosquitoes (*Zohdy et al., 2016*), which we identified only to genus level in the field. We recorded several minutes of acoustic data from both, to form a frequency distribution reference database specific to the RNM field site, for comparing new data from phone recordings (*Figure 4A,B*). We obtained two clearly separated frequency distributions with a single peak each, implying that the two genera were easily distinguishable by wingbeat frequency alone. About 60 recordings were collected in a 3-hour period by 10 volunteers who were trained for about 15 minutes and stationed at specific locations in the village. This number of recordings exceeded the number of mosquitoes simultaneously captured in CDC light traps placed at the same locations. Mapping the distributions of the two genera of mosquitoes across the village revealed considerable heterogeneity in the proportions of *Anopheles* and *Culex* mosquitoes from the western riverside to the eastern hillside, possibly influenced by extremely local factors such as drainage, density of humans, or presence of livestock (*Figure 4C*). This heterogeneity was concordant with the data from CDC light traps, which collected *Anopheles* mosquitoes only in those locations where they were also recorded by volunteers (*Figure 4C*, insets). Although the ratio of the two kinds of mosquitoes varied between acoustic recording and conventional trapping at each location, there was a qualitative correspondence between the two methods in terms of the relative number of mosquitoes at a location, and the more numerous species.

Together, the two pilot field trials at BBR and RNM highlight spatial variations in mosquito distributions along with their circadian activity patterns, in very different field environments. The greater number of observations obtained by pooling mobile phone recordings as compared to trapping indicates that phones may be a more productive means of sampling mosquito populations. At the same time, the maps of mosquito activity at both field sites were extremely localized, on the level of minutes and tens of metres. This demonstrates how mobile phone based crowdsourcing can simultaneously enhance both the scale and the resolution of ecological measurements of mosquito populations, indicating considerable potential for such an approach to the spatiotemporal mapping of mosquito vectors.

## Discussion

In this work, we demonstrate a new technique for mosquito surveillance, using mobile phones as networked acoustic sensors to collect wingbeat frequency data for automated identification. We present quantitative analyses of mobile phone acoustic signal quality and the differentiability of

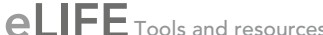

**Figure 4.** Spatio-temporal activity of mosquitoes in the field can be mapped using acoustic data collected by mobile phone users. (A) Sample spectrograms from female *Culex spp.* (top) and *Anopheles spp.* (bottom) mosquitoes captured in the field at Ranomafana in Madagascar. (B) Frequency distributions for field-caught *Culex spp.* and *Anopheles spp.* mosquitoes in Ranomafana, forming a reference for identification of recordings from either species at this field site. Acoustic data were collected for 3 minutes each, from 50 individual *Culex* and 10 individual *Anopheles* mosquitoes. (C) Map of Ranomafana village showing distribution of female *Culex spp.*, *Anopheles spp.*, and *Mansonia spp.* mosquitoes, from mobile phone data recorded by 10 volunteers over the approximately 1 km X 2 km area. Each square represents one recording, and black circles indicate locations where volunteers reported encountering no mosquitoes. The numbers in the white boxes show the number of *Culex* (pink) and *Anopheles* (gray) mosquitoes

*Figure 4 continued on next page*

*Figure 4 continued*

captured in CDC light traps over the same time period at those locations. The map shows a spatial gradient from riverbank to hillside in the relative proportion of *Anopheles spp.* and *Culex spp.* mosquitoes. Further, mosquito hotspots are interspersed with points having a reported lack of mosquitoes, highlighting the potential importance of factors such as the distribution of water and livestock. (D) Spatio-temporal activity map for female *Ae. sierrensis* mosquitoes in the Big Basin Park field site, using data collected by 15 hikers recording mosquitoes with their personal mobile phones, over a 3-hour period in an approximately 4.5 km X 5.5 km area. Each brown square represents one *Ae. sierrensis* female recording, and black dots represent sites where hikers reported encountering no mosquitoes at all. (Inset top left) Temporal distribution of the overall mosquito activity data depicted in (D) based on recording timestamps, showing the rise and fall in the number of recordings made, a proxy for mosquito activity, in each hour of the field study.

DOI: https://doi.org/10.7554/eLife.27854.017

The following figure supplements are available for figure 4:

**Figure supplement 1.** Mobile phones are capable of acquiring mosquito sounds in a variety of field environments.

DOI: https://doi.org/10.7554/eLife.27854.018

**Figure supplement 2.** Individual flight traces for wild mosquitoes show highly similar mean frequencies with small but intrinsic variances.

DOI: https://doi.org/10.7554/eLife.27854.019

important mosquito species, complemented by field data collected by minimally trained volunteers and organized into spatio-temporal maps.

The involvement of local volunteers in our study underlines that practically anyone with a mobile phone can quickly be trained to contribute data towards mosquito surveillance efforts. Based on the proof-of-concept presented here for a mobile phone based surveillance framework, we propose a citizen science effort for mapping mosquito populations (*Figure 1—figure supplements 1* and *2*). Citizen science is an exciting paradigm for community-based vector surveillance, with emerging initiatives such as the United Nations-backed Global Mosquito Alert. These programs rely on data collected by volunteers from the general public, in the form of physical specimens or through apps for reporting the presence of mosquitoes with photographs for expert identification. Our proposed platform would support the collection of mosquito sound recordings from mobile phone users, automating the audio processing to instantly identify and map the species without human intervention. This approach has the potential to greatly expand surveillance efforts in resource-limited areas where they are needed the most, by enabling people to take the initiative in tracking disease vectors within their own communities.

The greatly reduced cost of surveillance as a result of using pre-existing infrastructure, together with the ability to localize mosquito recordings, allows both the coverage and resolution of mosquito field maps to increase. The availability of sufficient data on highly local scales could inform mosquito control strategies tailored to a given location. Further, this approach also has the potential to screen for the presence of exotic or invasive species, particularly in high-risk areas such as port cities. Additionally, it provides a rapid and easy way to survey baseline mosquito populations in a given area and their fluctuations over time, which may be highly relevant for programs that aim to control mosquito populations through the release of sterile or genetically modified organisms. The implicit need for human involvement in our method skews the collection strongly towards species that are most likely to encounter a human. Although all methods of mosquito collection, including various traps, have biases in the species they predominantly attract, the over-representation of anthropophilic species is a caveat to be noted when probing mosquito ecology using our method.

We have focused on statistical Maximum Likelihood Estimation methods for acoustic identification of mosquitoes, using known distributions corresponding to different species as a reference. There are however many possible methods to predict the species that a given recording belongs to, with various techniques for incorporating prior knowledge about the wingbeat frequency, circadian activity, habitat and other characteristics of the relevant mosquito species. The field of computational entomology, which includes the development of algorithms for the automated identification of mosquito sound in acoustic data and its classification by species is an active area of research, with growing relevance for the automation of mosquito surveillance (*Chen et al., 2014*; *Silva et al., 2015*; *Kiskin et al., 2017*). There is considerable scope for studies that not only search for more acoustic features that discriminate between mosquito species, but also those which use spatiotemporal population data to identify trends and patterns. These can be correlated to other man-made and environmental factors, to enable evidence-based decision making for mosquito control and

disease management. Such insights require not only the development of algorithms capable of sifting through surveillance data, but also considerable expansion of our databases of prior knowledge to train these algorithms to identify more species across diverse areas of the world.

Recent technological trends have spurred a rapid rise in apps and services based on crowdsourcing data about the physical world through embedded sensors and user input from mobile phones. Applying analytics methods and spatiotemporal mapping on this data provides useful information ranging from traffic density to restaurant recommendations to earthquake warnings. At present, the critical missing link in enabling similar advances in mosquito surveillance and control is the capacity to generate large quantities of mosquito ecological data on fine-grained space and time scales (*Ferguson et al., 2010*). Our mobile phone based solution holds great promise as a scalable, non-invasive, high-throughput and low-cost strategy to generate such data, by leveraging widely available hardware and an existing network infrastructure. We hope that a citizen science approach to mosquito surveillance based on this method will boost our capability to dynamically assess mosquito populations, study their connections to human and environmental factors, and develop highly localized strategies for pre-emptive mosquito control (*Hay et al., 2013*).

## Methods and materials

### Acquisition of acoustic data from mobile phones

Different mobile phones were used to collect sounds from mosquitoes, using audio recording software that was already freely available on the devices, including applications for voice messaging, voice memos, or sound recording. To locate the primary microphone on the body of the mobile phone, we either read the location of a user manual showing its different components, or found the microphone by trial-and-error, where we tapped the phone periphery and observed the response of the recording software to locate the area with maximum audio sensitivity. The primary microphone is oriented towards the mosquito for maximum sensitivity in audio acquisition. Audio data from the phones was compiled and transferred to a server for processing. This method applies to all figures.

### Processing of acoustic data from mobile phones

The audio signals from mosquitoes were acquired at different sampling rates ranging from 8 kHz to 44.1 kHz and a variety of file formats, depending on the mobile phones and the specific in-built or user-defined settings on the recording applications used to acquire the signal. When a raw signal was acquired from the phone, it was converted into the WAV format for convenience of processing, with sampling rate interpolated to 44.1 kHz if sampled at a lower rate. To reduce constant background noise in the signal, we used a spectral subtraction algorithm, with the fundamental principle of subtracting the actual or expected frequency content of pure noise from the spectrum of the noisy signal. Here, we identified the background using an automated algorithm as those spectral bands that are constant with almost zero variation in amplitude and frequency across the entire sound clip. To generate the audio frequency spectrum over different instants of time, we applied the Short Time Fourier Transform (STFT ; Signal Processing Toolbox, MATLAB R2015B) to produce a spectrogram with resolutions of 5 Hz in frequency and 20 ms in time, having a high degree (90%) of overlap between windows to achieve a trade-off between sufficient frequency resolution and accurate localization of the signal in time.

### Construction of wingbeat frequency distributions (*Mukundarajan et al., 2017*)

Using the spectrogram, we construct histograms for the distribution of peak frequencies for mosquitoes of a given species. Once a sound was identified (either manually or using an automated code) as belonging to a mosquito of a given species, the lowest frequency corresponding to a local maximum in amplitude was detected using a peak finding algorithm. In cases where more than one mosquito is recorded sequentially in the same file, they are considered separate instances, distinguished as contiguous durations of peak frequency in a given range separated by time. In cases where recordings from mosquitoes overlap, the louder mosquito is considered. In many studies, a representative wingbeat frequency is found for a single flight trace by averaging the fundamental frequency over all time instants. However, the wingbeat frequency from a single continuous mosquito

signal can vary over a range of about 100 Hz for a tethered mosquito, and over a few tens of Hz for mosquitoes in free-flight. To take into account this natural variation in wingbeat frequency over the duration for an audio signal, we did not average the frequency over time, but instead focused on the instantaneous frequency computed in each 20 ms interval time window. Each time window was treated as an individual instance, and its fundamental frequency added to a histogram, to bring out the natural variations in wingbeat frequency within a single signal. Peak frequencies from all time windows are binned with a bin size of 5 Hz - the same frequency resolution imposed upon the spectrogram. This yields a histogram that captures the distribution of peak frequencies for that mosquito species, without making *a priori* assumptions about the nature of this distribution. For statistical computations, the histograms are normalized by the total number of instances such that the area under the distribution sums up to 1, yielding a discrete probability density distribution for wingbeat frequency. However, for ease of representation in figures, probability mass functions where the histogram counts are normalized relative to the maximum number of counts in a single bin are shown, so that each histogram spans 0 to 1 on the y-axis. This method applies to all figures.

## Statistical tests and metrics

Wingbeat frequency was represented as discrete probability density distributions, with the frequency binned into intervals of 5 Hz (the computational accuracy for frequency in our STFT analyses) and the area under the probability density distribution summing up to 1. The peak frequency measured in each time window of the video or audio was treated as an individual sample, and the 2-sample T-test carried out using the MATLAB Statistics toolbox at a significance level of 1% for peak frequencies of the 165 time windows compared. The peak frequency samples measured from video and audio were tested to check if the distributions had highly similar mean and variance, indicating that they have been sampled from the same probability distribution function. This method applies to *Figure 2C*. The Bhattacharya overlap coefficient (BC) and Jensen-Shannon Divergence metric (JSD) were computed between pairs of wingbeat frequency distributions, where the number of counts of wingbeat frequency in each bin are normalized by the product of the total number of samples and the range of the independent variable (frequency, 200 to 700 Hz) to yield a probability density distribution with unit area under the curve. BC is calculated as the sum of the geometric mean of the two probability densities in each bin, ranging between a minimum of 0 for disparate non-overlapping distributions and a maximum of 1 for identical distributions. The JSD is calculated as the square root of the arithmetic mean of the Kullback-Leibler divergences of each distribution with respect to the other, for each pair of distributions considered. These methods apply to *Figure 2G,H*, and *Figure 3B*.

## Testing and validation of Maximum Likelihood Estimation (MLE) algorithm

### Algorithm description

In this classification algorithm, the different species are treated as discrete values of a parameter that alters the point mass function for the distribution of frequency from mosquito sound. A given observation is statistically compared to each distribution to deduce which one it is most likely to have been sampled from, and the species with maximum likelihood to be the parent distribution is identified. We chose this strategy as an appropriate method for identification, as it attempts to identify the statistical distribution that is most likely to have produced the observed data, i.e. it attempts to identify the species that is most likely to have produced an observed sound recording. This is particularly appropriate as we treat each recording not as a single data point, but as a distribution of wingbeat frequencies over the duration of the recording. Hence, to compare and classify the observed distribution from the recording with respect to the various reference distributions, MLE provides a natural solution. We use wingbeat frequency data from each species, isolated for each 20 ms sample window using a peak finding routine on the spectrogram as described earlier, to construct the point mass functions that describe the wingbeat frequency distributions for each species. We also treat any given acoustic recording being queried as a collection of discrete samples of wingbeat frequency from 20 ms windows. For each sample window frequency in this recording, we find the probability density of its occurrence in the frequency distributions corresponding to each species. We thus construct a matrix that gives the probability of occurrence in each of the frequency

distributions corresponding to each species, for each sample point of frequency in the given recording. Assuming each sample point of frequency in the given recording to be independent, we now sum across all sample points to obtain a measure of the combined likelihood that all sample window frequencies in the recording were produced from each species frequency distribution. We add these values instead of multiplying, to avoid the case where a single outlier frequency value in a recording has the power to veto a certain species due to having zero probability of occurrence in the corresponding frequency distribution. We choose the species which has the maximum likelihood for the total over all sample windows of that recording, as the species with the maximum likelihood of having produced a recording with that given set of frequency values.

### Location filter
We constructed a location filter by looking up the presence or absence of all the species in our database in a set of 200 countries, as recorded in the Walter Reed Biosystematics Unit Mosquito Identification Database (*Walter Reed Biosystematics Unit, 2017*). When applying the filter, we choose a location at random for a recording from a given species, based on its geographical distribution as specified in our location matrix. When finding the species with the maximum likelihood of having produced the frequencies observed in that recording, we now only choose from other species that are also located in the country chosen for our recording, as per our location matrix. This winnows down the number of choices for the species, eliminating those with similar frequency distributions but which are not geographically relevant.

### Validation using bootstrapped data
We first applied our algorithm to find the probability of correct classification of bootstrapped data from the original set of frequency values that was used to generate the wingbeat frequency distributions for each species. This is expected to correctly classify species whose frequency distributions do not have much overlap with others, and make mistakes when classifying species which have significant overlaps with others. For each species, we choose a random subset of frequency values from the original distribution and assign it to the species with maximum estimated likelihood. We repeat this for a total of 10,000 random subsets per species, to find the asymptotically convergent probability of classification for a large number of trials. A confusion matrix was generated by finding the fractions of these 10,000 trials for each species that were classified correctly, or erroneously to a certain other species. As expected, we see very high classification accuracies (greater than 0.99) for species with relatively non-overlapping frequency distributions, with errors arising exclusively in the classification of overlapping species.

### K-fold validation by partitioning data
To get an idea of classification errors that arise when the data being queried is not itself used to generate the distribution against which it is being compared, we performed K-fold validation for K from 2 to 10. Here, we partitioned the data into K groups, and used one group to generate queries which were matched against distributions generated using the other (K-1) folds. The confusion matrix generated in this case had very similar values as the one generated for bootstrapped data.

### Validation using test data
We finally validated our approach against test data that was collected in a separate experiment using a different phone, from the same reference populations. Contiguous individual mosquito traces were chosen from these recordings, and compared against the reference distributions. This validation was performed both with and without the location filter, to show the effect of including location metadata to refine classification.

## Mosquito specimens
Mosquito colonies were sourced from a number of different labs and facilities, including our own. Mosquitoes were typically chosen to be between 5 and 15 days, with all individuals in a colony aged within two days of each other. Females were typically mated but not bloodfed. The lab reared colonies used in this work sourced from the BEI Resources Vector Resource collection were *Aedes aegypti* (strain COSTA RICA, provided by the Animal Flight Lab at UC Berkeley), *Aedes aegypti*

(strain NEW ORLEANS, reared at CDC, Atlanta), *Aedes aegypti* healthy and infected with *D. immitis* (strain NEW ORLEANS, provided by the Zohdy Lab at Auburn University), *Aedes albopictus* (strain ALBOPICTUS, reared at CDC, Atlanta), *Anopheles albimanus* (strain STECLA, reared at CDC, Atlanta), *Anopheles arabiensis* (strain DONGOLA, reared at CDC, Atlanta), *Anopheles arabiensis* (strain RUFISQUE, reared at CDC, Atlanta), *Anopheles atroparvus* (strain EBRO, reared at CDC, Atlanta), *Anopheles dirus* (strain WRAIR2, reared at CDC, Atlanta), *Anopheles farauti* (strain FAR1, reared at CDC, Atlanta), *Anopheles freeborni* (strain F1, reared at CDC, Atlanta), *Anopheles gambiae* (strain KISUMU, reared at CDC, Atlanta), *Anopheles gambiae* (strain AKRON - bendiocarb resistant, reared at CDC, Atlanta), *Anopheles gambiae* (strain RSP - permethrin resistant, reared at CDC, Atlanta), *Anopheles merus* (strain MAF, reared at CDC, Atlanta), *Anopheles minimus* (strain MINIMUS1, reared at CDC, Atlanta), *Anopheles quadriannulatus* (strain SANQUA, reared at CDC, Atlanta), *Anopheles stephensi* (strain STE2, reared at CDC, Atlanta), *Anopheles stephensi* (strain STE2, provided by the Luckhart Lab at UC Davis), *Culex quinquefasciatus* (strain JHB, reared at CDC, Atlanta), *Culex tarsalis* (strain Yolo, reared by us).

Colonies caught in the field or bred from catches included *Aedes aegypti* (F1, Los Angeles, provided by the Coffey Lab at UC Davis), *Aedes aegypti* (F1, Puerto Rico, provided by the Coffey Lab at UC Davis), *Aedes mediovittatus* (F0, provided by the Coffey Lab at UC Davis), *Anopheles quadrimaculatus* (F22, Alabama, provided by the Mathias Lab at Auburn University), *Culex pipiens pipiens* (provided by the Santa Clara Vector Control District), *Culex pipiens pipiens* (provided by the San Mateo Vector Control District), *Culex quinquefasciatus* (provided by the San Mateo Vector Control District). Wild mosquitoes captured by us in field trials include *Culiseta incidens* (captured at Jasper Ridge Biological Preserve, Stanford University, and in San Francisco), *Ochlerotatus sierrensis* (captured at Big Basin Redwoods State Park, California, USA), *Anopheles spp.*, *Culex spp.* and *Mansonia spp.* (captured at the Centre ValBio and Ranomafana village, Madagascar).

## Comparisons with high speed videography

We acquired high speed video of tethered mosquitoes in the lab using a Phantom v1610 camera, at 10,000 frames per second (*Video 5*). Simultaneously, we made audio recordings using a mobile phone placed with the primary microphone 10 mm away and oriented towards the mosquito. Since the audio and video are completely independent as the camera does not talk to the phone, synchronization was achieved using a specially designed setup to produce a specific light and sound pattern. We connected a piezoelectric buzzer and an LED to the same pin of an Arduino. The buzzer and LED are both fed the same pattern of voltage by the microprocessor. Hence, the frequency of sound from the buzzer recorded in the audio exactly matches the frequency of light flashes from the LED recorded in the video at every instant of time, allowing the audio and video spectrograms to be aligned in time. We programmed the Arduino to produce a square wave at 5000 Hz and 50% duty cycle for 500 ms, followed by a 500 ms pause, and then a square wave at 2000 Hz for 500 ms. This gave us four time points - the beginning and end of each waveform - to use for aligning the corresponding spectrograms from video and audio in time. The video data were thresholded and the area of the wing (which we recorded face on) was computed in each frame. We plotted a waveform of the change in projected wing area over time, and applied the STFT to produce a spectrogram. The fundamental frequency in the spectrogram corresponded to the wingbeat, with higher harmonics corresponding to subtle variations in wing kinematics such as wing deformation during clap-and-fling. Acoustic data from the mobile phone was processed as described in the section above. The two spectrograms were computed to the same time and frequency resolutions of 5 Hz and 20 ms, and aligned in time based on the best match of the four points of synchronization. This method applies to *Figure 2B*.

## Comparisons to studio microphones

Comparison with an acoustic gold standard was achieved using the Marshall MXL991 and Apex 220 microphones, the latter of which is calibrated to have a flat frequency response between 100 and 1000 Hz. The two studio microphones were connected to a pre-amplifier (Onyx) with the gain set to its maximum value of 60 dB, after ensuring that this would still avoid saturation. We carried out experiments to calibrate the sensitivity of mobile phones over distance using a standardized sound source - a piezoelectric buzzer ringing at 500 Hz, with its amplitude measured before every

recording to be constant at 77 dB at the edge of the buzzer disc. To compare the ability of mobile phones to record mosquito sounds, we recorded tethered male and female *Culex tarsalis* mosquitoes. We placed the microphones at an identical distance to the left of the mosquito as we placed the mobile phone primary microphone to its right, since waveforms produced by the two wings are assumed to be symmetrical. We synchronized recordings from all three sources using the times of initiation and cessation of wingbeat sound, with multiple flight traces in a single dataset. Using the amplitudes recorded by the Marshall MXL991 studio microphone which has a known $1/r^2$ drop in recorded amplitude, we deduced the actual amplitude produced by the mosquito in a given experiment, and standardized the corresponding mobile phone SNR for a uniform source amplitude of 45 dB (which we measured to be a typical amplitude produced by a mosquito). Spectrograms constructed independently and aligned in time are shown in *Figure 1—figure supplement 1*. These methods apply to *Figure 2D,E* and *Figure 1—figure supplement 1*.

## Acoustic data collection in the lab from tethered mosquitoes

Individual sound traces for distance calibration experiments were collected from tethered mosquitoes. Individuals were aspirated out of the cage and knocked out with a puff of carbon dioxide. The wings were gently spread to move them out of the way, and a pipette tip was affixed to the scutum with a bead of low-melting insect wax. The pipette tip was clamped in a stand, and the appropriate recording device - mobile phone, studio microphone or high speed camera - was clamped in another stand at the desired orientation and a specified distance away from the pipette tip as measured by a ruler. The legs of the mosquito are gently stimulated by touch to induce a flight reflex, after which the wings beat for a period of a few seconds to minutes. This method applies to *Figure 2A,B,D,E*.

## Acoustic data collection in the lab from caged populations

Wingbeat frequency distributions for a given species were measured from lab-reared populations maintained in 1-ft cubical cages. Cages typically contained between 100 and 300 individuals of males and females each, with the sexes segregated into separate cages whenever possible. The mobile phones were inserted by hand through the stockinette sleeves of the cages, with the primary microphones oriented away from the hand, and moved to follow individual mosquitoes in flight or against walls or corners of the cage. Care was taken to avoid introducing noise from bumping against the cage surfaces or rubbing against the sleeve. Between 5 and 10 minutes of data were collected per cage, and high amplitude noise due to bumps was eliminated using an automated algorithm. This method applies to *Figure 2F*, *Figure 3A* and *Figure 1—figure supplement 2*. In some cases, individual mosquitoes were introduced into an otherwise empty cage, to record free flight traces from a specific mosquito in the lab. This method applies to *Figure 4A,B*.

## Acoustic data collection in the field

Field acoustic data were collected in a variety of locations, including homes and gardens around San Francisco and Palo Alto, USA, at Stanford University's Jasper Ridge Biological Preserve, and around the Centre ValBio in Ranomafana, Madagascar. Mosquitoes were either followed with a phone during free flight around the user, or captured live in a Ziploc bag and subsequently recorded by putting the phone's primary microphone against the bag (while taking care not to introduce noise due to crumpling or brushing the bag surface). This method applies to *Figure 4*. The pilot demonstrations of field recording and mapping of mosquitoes were organized with small teams of volunteers working in pairs, with 14 and 10 users respectively in Big Basin Redwoods State Park in California, USA (20 km², between 5:30PM and 8:30PM on 17 August 2016), and in Ranomafana village, district Ifanadiana, Madagascar (4 km², between 6PM and 8PM on 26-28 October 2016). During the studies, volunteers were hiking along trails in Big Basin Park, and were gathered in houses or shops in Ranomafana village. Prior to the field study, we initially collected live mosquitoes from the field, recorded them in the lab to create a curated database of signatures for those specific locations, and later morphologically identified them through microscopy for association with each acoustic signature (*Figure 4A,B*). Subsequently, field recordings were made by the teams, and each recording was assigned a species by comparing with the databases (*Figure 4B*). During the field exercises, the users also collected matched physical specimens from the field in grinder tubes and Ziploc bags corresponding to many of the audio recordings, which were morphologically identified to confirm the

IDs assigned based on the acoustic database. The recordings were associated with a location as reported by GPS or the user, and timestamped automatically by the recording application on the mobile phone, for spatio-temporal placement of each observation. Maps were prepared by counting the number of reliably identified acoustic signatures for each location. This method applies to *Figure 4C,D*.

### Downloadable data package

As part of our policy to openly share all data from this project, we have included a downloadable package comprising all acoustic data collected over the course of this work. This includes acoustic recordings from 20 different species of mosquitoes, using a variety of mobile phones for each. This data can be downloaded either from our project website at abuzz.stanford.edu, or from the online repository at Dryad (http://dx.doi.org/10.5061/dryad.98d7s). The supplementary audio files are not included in this package, and may be downloaded separately.

## Acknowledgements

We acknowledge Amalia Hadjitheodorou, members of the Prakash Lab, and our data collection volunteers for assistance in field experiments. We thank Bernadette Rabaoarivola (Menja) and Malalatiana Rasoazanany for their invaluable participation in field studies in Madagascar, and also Mainak Chowdhury and Saptarshi Bandyopadhyay for their valuable inputs regarding the analysis of our data. We also thank Rebecca Konte for designing graphical handouts and posters describing mosquito recording techniques for public outreach. We are grateful to Ellen Dotson and Mark Benedict at the US Centers for Disease Control, the Luckhart and Coffey labs at UC Davis, the Zohdy and Mathias labs at Auburn University, and the Animal Flight Lab at UC Berkeley, for providing us with access to their mosquito colonies for collecting audio data. We further acknowledge Big Basin Redwoods State Park, California, USA, where we collected acoustic signatures of mosquitoes on hiking trails. We are thankful to the Centre ValBio in Ranomafana, Madagascar for enabling our field studies in the area, and to PIVOT and particularly Andrés Garchitorena for coordinating our field based efforts. We are grateful to Nona Chiariello and the Jasper Ridge Biological Preserve at Stanford University, the Santa Clara County Vector Control District and the San Mateo County Mosquito Control District for their support and assistance with providing access for field work, materials for cultures, and assisting with identification of field specimens. The materials for cultures in our lab were provided by BEI Resources and NIAID. Finally, we thank Desirée LaBeaud, Erin Mordecai, John Dabiri, and all members of the Prakash Lab for illuminating discussions about our work and comments on our manuscript.

HM acknowledges support from a HHMI International Student Research Fellowship, EAC from a Stanford Mechanical Engineering Graduate Fellowship, and FH from a NWO Rubicon Fellowship. MP acknowledges support from the NSF Career Award, HHMI-Gates Faculty Fellows program, the Pew Foundation Fellowship and the MacArthur Fellowship. This work was additionally supported by the Coulter Foundation, the Woods Environmental Institute at Stanford University, the NIH New Innovator Grant and the USAID Grand Challenges: Zika and Future Threats award.

## Additional information

### Funding

| Funder | Grant reference number | Author |
|---|---|---|
| Howard Hughes Medical Institute | International Student Research Fellowship | Haripriya Mukundarajan |
| Nederlandse Organisatie voor Wetenschappelijk Onderzoek | Rubicon Postdoctoral Fellowship | Felix Jan Hein Hol |
| National Science Foundation | | Manu Prakash |
| Howard Hughes Medical Institute | | Manu Prakash |
| NIH Office of the Director | | Manu Prakash |

| Pew Charitable Trusts | Manu Prakash |
| John D. and Catherine T. Ma-cArthur Foundation | Manu Prakash |
| United States Agency for In-ternational Development | Manu Prakash |

The funders had no role in study design, data collection and interpretation, or the decision to submit the work for publication.

## Author contributions
Haripriya Mukundarajan, Conceptualization, Data curation, Formal analysis, Validation, Investigation, Methodology, Writing—original draft, Writing—review and editing; Felix Jan Hein Hol, Formal analysis, Validation, Investigation, Methodology, Writing—original draft, Writing—review and editing; Erica Araceli Castillo, Formal analysis, Investigation, Methodology, Writing—review and editing; Cooper Newby, Conceptualization, Investigation, Methodology, Writing—review and editing; Manu Prakash, Conceptualization, Supervision, Funding acquisition, Investigation, Methodology, Writing—original draft, Writing—review and editing

## Author ORCIDs
Haripriya Mukundarajan (iD) http://orcid.org/0000-0001-8220-0724
Manu Prakash (iD) http://orcid.org/0000-0002-8046-8388

## Decision letter and Author response
Decision letter https://doi.org/10.7554/eLife.27854.033
Author response https://doi.org/10.7554/eLife.27854.034

## Additional files

### Supplementary files
• Supplementary file 1. Sample acoustic recording from a lab-reared 5-day-old female *Anopheles gambiae* mosquito in a cage at the CDC insectary, Atlanta, GA, USA. Recording was done by one of the authors, using a 2006-model Samsung SGH T-209 clamshell phone sampling at 8 kHz.
DOI: https://doi.org/10.7554/eLife.27854.020

• Supplementary file 2. Acoustic recording from wild male *Culex pipiens* mosquito in a living room in Menlo Park, CA, USA. Recording was done by one of the authors using a 2010-model iPhone 4, sampling at 48 kHz.
DOI: https://doi.org/10.7554/eLife.27854.021

• Supplementary file 3. Acoustic recording from wild female *Culex pipiens* mosquito in a bathroom in Redwood City, CA, USA. Recording was done by a volunteer using a 2014-model Samsung Galaxy A3, sampling at 44.1 kHz.
DOI: https://doi.org/10.7554/eLife.27854.022

• Supplementary file 4. Acoustic recording from wild female *Culex pipiens* mosquito in a garden in Menlo Park, CA, USA. Recording was done by one of the authors using a 2010-model iPhone 4, sampling at 48 kHz.
DOI: https://doi.org/10.7554/eLife.27854.023

• Supplementary file 5. Acoustic recording from wild female *Culiseta incidens* mosquito in a garden in San Francisco, CA, USA. Recording was done by one of the authors using a 2015-model iPhone 6S sampling at 48 kHz.
DOI: https://doi.org/10.7554/eLife.27854.024

• Supplementary file 6. Acoustic recording from wild female *Aedes sierrensis* mosquito at Big Basin Redwoods State Park, CA, USA. Recording was done by a volunteer using a 2015-model Sony Xperia Z3 compact, sampling at 44.1 kHz.
DOI: https://doi.org/10.7554/eLife.27854.025

• Supplementary file 7. Acoustic recording from wild female *Culiseta incidens* mosquito near a highway at Jasper Ridge Biological Preserve, CA, USA. Recording was done by one of the authors using a 2006-model Samsung SGH T-209 clamshell phone sampling at 8 kHz.
DOI: https://doi.org/10.7554/eLife.27854.026

• Transparent reporting form
DOI: https://doi.org/10.7554/eLife.27854.027

• Supplementary file 8. Acoustic recording from wild female *Anopheles spp.* mosquito near an outdoor pig pen in Ranomafana, Madagascar. Recording was done by a volunteer using a locally available non-smart phone sampling at 44.1 kHz.
DOI: https://doi.org/10.7554/eLife.27854.028

• Supplementary file 9. Acoustic recording from wild female *Culex spp.* mosquito inside a local residence in Ranomafana, Madagascar. Recording was done by a volunteer using a locally available non-smart phone sampling at 44.1 kHz.
DOI: https://doi.org/10.7554/eLife.27854.029

• Supplementary file 10. Acoustic recording from wild female *Aedes sierrensis* mosquito at Big Basin Redwoods State Park, CA, USA. Recording was done by a volunteer using a 2015-model HTC-One M8 phone sampling at 44.1 kHz.
DOI: https://doi.org/10.7554/eLife.27854.030

## Major datasets

The following dataset was generated:

| Author(s) | Year | Dataset title | Dataset URL | Database, license, and accessibility information |
| --- | --- | --- | --- | --- |
| Haripriya Mukundarajan, Felix Jan Hein Hol, Erica Araceli Castillo, Cooper Newby, Manu Prakash | 2017 | Data from: Using mobile phones as acoustic sensors for mosquito surveillance | http://dx.doi.org/10.5061/dryad.98d7s | Available at Dryad Digital Repository under a CC0 Public Domain Dedication |

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
