## [Decision Letter]

Thank you for submitting your article "Using mobile phones as acoustic sensors for high-throughput surveillance of mosquito ecology" for consideration by *eLife*. Your article has been favorably evaluated by Prabhat Jha (Senior Editor) and five reviewers, one of whom serve as Guest Reviewing Editor. The following individual involved in review of your submission has agreed to reveal their identity: Scott Ritchie (Reviewer #5).

The reviewers have discussed the reviews with one another and the Reviewing Editor has drafted this decision to help you prepare a revised submission.

Summary:

This work presents an evaluation of the use of common mobile phones for mosquito detection (acoustically), attempts to differentiate species via their acoustic signatures alongside other meta data. Overall this is a very interesting effort that provides an out-of-the-box approach to mosquito and vector surveillance. The reviewers expressed excitement and need for this approach, note that the authors provided technical detail and a convincing case. However there are several things that could be done in order to make a better case for the feasibility of this approach, which are outlined here as essential revisions.

Essential revisions:

A) Differentiating species: Overall the reviewers noted the need to provide more information or demonstration this can accurately be done. Details towards this include:

-Provide more details on meta data required for differentiation (e.g. biting times, geographic prevalence data (and especially in Madagascar), how do you handle multiple counts… (mosquitoes may be in swarms and same one could be counted many times), discussion of why are some distributions not discernible, how many counts needed to generate a distribution that is discernible (e.g. Figure 4), and/or what total bandwidth (flight time) needed?

-Along the same lines, reviewers concurred that results as presented do not really demonstrate that given a single measurement the species can be differentiated. Reviewers suggested that perhaps the authors could frame the question differently, instead of: given that the mosquito belongs to species A or B, can we work out which? The problem with this is that of course there are multiple possible species a sample could belong to. Presumably a single measurement would not be able to demonstrate which species a sample came from, but if they record the mosquito for long enough, can the species be determined by comparing the distribution of measurements against their reference bank?

-Also reviewers wondered: with the wild caught Anopheles mosquitoes in Madagascar was it not possible to differentiate the species? Much is made earlier in the text of being able to differentiate members of the *Anopheles gambiae* complex yet this isn't done in the wild. Was this because species were not identified in the field or because they could not be distinguished? Either way this should be highlighted.

-Discussion of how does the inter-species variability relate to the intra-species variability. Comparison of wild caught mosquitoes shows that the inter-quartile range is substantially different (non-overlapping) and varies between different sized individuals in the same population. The magnitude of this variation seems greater than the difference between species the authors say it can differentiate though this is not mentioned. Comparison of the wild and colony mosquitoes is not done and should be discussed.

-One reviewer also noted that the statement that *Aedes aegypti* and *Anopheles gambiae* mosquitoes can be differentiated by their biting time is wrong. "To my knowledge though *An. gambiae* are unlikely to bite in the day both are highly active at dawn and dusk and *Aedes* can certainly be found biting at night. Without the use of this meta-data these mosquitoes could not be distinguished acoustically and this should be highlighted."

-Inter vs. Intra colony and variability: The Materials and methods shows multiple colony mosquitoes of each species though sometimes these appeared grouped together in Results (e.g. *Anopheles arabiensis*) though at other times they were separated (e.g. *An. quadrimaculatus*, Figure 3). Nowhere in the manuscript is inter-colony variation mentioned and it needs to be as it is immensely important. If new calibration datasets are needed for each mosquito sub-population the potential for citizen science is massively diminished. The inter-colony, intra-colony and inter-individual variability all needs to be discussed and ideally separated as currently they are all grouped together making it impossible for the reader to determine the accuracy of the method.

B) Classification details and methods: the reviewers suggest the authors consider advanced methods to improve classification (e.g. machine learning) *and* provide more details on frequency distributions and measurement requirements to around the classification and classification accuracy. Details:

-Classification accuracy (albeit unblended) could and should be presented more quantitatively for the data presented in Figure 3 – at least at the pairwise level (proportion of individuals correctly classified in a mix of 2 species). A more sophisticated analysis might be to sample potentially collocated species from the data in 3A and present multi (>2) species classification accuracy.

-Machine learning methods should be considered to optimise classification accuracy.

-One reviewer also noted that it was not clear why the higher harmonics/overtones (Figure 1, Figure 2) were being thrown away in the analysis presented. Even simple random forest methods trained on relative amplitude/frequency pairs (of course other features could be selected, such as measures of spread across a trace) for the dominant frequency and harmonics might offer considerably greater classification accuracy than just the mean dominant frequency.

C) Experiments: The reviewers suggested some experiments to demonstrate that it is possible to discern relative abundance would be helpful. While time may not permit a systematic field study, we offer the following strongly suggested approach for a blinded test (please note it is not the reviewer's intent for these experiment(s) to take longer than the revision time, if the authors do not think these are feasible in the allotted time period please justify why):

-Replicate experiments where the authors mix (at known proportions) individuals from several vector species in an experimental large cage setting, and then get researched blinded to the species mix to (a) sample wing beat frequencies from multiple (or all) insects, and (b) analyse the results to determined predictive/measurement accuracy. If mixing mosquitoes is too hard, then sequential recordings of a mix of mosquitoes in the same environment and blinded analysis might suffice.

D) Range:

-The reviewers noted that given the limited range as described (5cm), it would be helpful to demonstrate that the approach works (or sensitivity of outcomes) if the orientation or distance between the microphone and mosquito are not ideal.

E) Data and accessibility:

-The authors are encouraged to follow open data norms and provide the raw data that went into the figures in the main manuscript as supplements to the paper.

-As well, in line with the aims of the paper the reviewers agreed that it would be great if the authors can elaborate on how Joe Public can actually get started with this (release data and app).

F) More discussion or real applications in terms of mosquito species and limitations:

-The reviewers note that this approach will work well for anthropophilic mosquitoes, but not as well as animal feeders esp. avian feeders that are not attracted to man.

-As well there is high applicability to ID exotics; the authors are encouraged to augment discussion of the approach's relevance to *Aedes* and, for example current citizen science program in Europe doing also surveys, or towards surveillance of albos in Australia (PLoS neglected tropical diseases, 11(2), p.e0005286)

-These include hard to catch and measure mosquitoes at night (when they are most active). Will size impact the WBF? Most wild mosquitoes are smaller than lab reared. Will this be a confounder to ID? And what about temperature and age?

-The reviewers noted that this would particularly be an appropriate tool to survey for exotic species, especially where the species of concern is known (e.g., *Ae. albopictus*). It would also be a great tool to measure populations for rear and release programs such as wolbachia and sterile males. In this instance, we need to know the wild population so that effective numbers can be released. Phones might be a great way to do this.

-*Oc. sierrensis* should be *Aedes sierrensis*.

-Also, how did they measure WBF? In a bag, or free flying?

-In several places you mention species-specific wingbeat frequencies. They are not really species specific as there is some overlap. Please remove species-specific.

---

## [Author Response]

Essential revisions:A) Differentiating species: Overall the reviewers noted the need to provide more information or demonstration this can accurately be done. Details towards this include:-Provide more details on meta data required for differentiation (e.g. biting times, geographic prevalence data (and especially in Madagascar), how do you handle multiple counts (mosquitoes may be in swarms and same one could be counted many times), discussion of why are some distributions not discernible, how many counts needed to generate a distribution that is discernible (e.g. Figure 4), and/or what total bandwidth (flight time) needed?

We have addressed these specific points directly in the main text, as described below.

Metadata needed for differentiation:

We have added a paragraph and new figures to the manuscript describing the importance of metadata in species classification (subsection “Acoustic data can be combined with metadata to improve classification accuracies”, first paragraph, Figure 3—figure supplement 1, Figure 3). Here, we describe the common types of metadata that can be collected by mobile phones (time stamps, geo-tags, photographs, behavior), We demonstrate that classification accuracy can be improved by adding metadata parameters, by constructing a filter based on geographic prevalence data. We discuss the inherent challenges in differentiating all mosquito species on wing beat frequency alone, and provide a direct demonstration of the value of metadata.

Apart from time, location, and images, there are several other parameters that can be useful in both species identification, and to study trends in mosquito behavior and human-mosquito interactions. These include weather conditions such as temperature, humidity and wind speed, whether the encounter was indoors or outdoors, in a public venue or otherwise, urban or wooded area, and in a swarm or in isolation. This provides a new stream of data that is extremely valuable for researchers. However, the association of these factors with specific mosquito species is not typically well understood and represents an avenue for future investigation.

We have clarified in the manuscript (subsection “Mobile phones can be effectively used to map mosquito activity through user-driven acoustic surveillance”, fourth and fifth paragraphs) that metadata collected in Big Basin and Madagascar was not used for differentiation of mosquitoes (which was solely based on wingbeat frequency), but was instead used for mapping out the mosquito data, as shown in Figure 4.

Handling multiple mosquitoes at a time:

We have included this detail in the Materials and methods section of our manuscript (subsection “Construction of wingbeat frequency distributions”). We clarify that when mosquitoes are recorded sequentially, each is treated as a different trace. When they are recorded simultaneously, our current algorithm selects the loudest one and interprets it as a single count. We do not explicitly account for the case of multiple counts of a single mosquito, since this is statistically equivalent to a longer recording of the same mosquito in our approach, where we treat flight traces as a collection of frequency values measured across sample windows rather than as a single instance.

Differentiability of distributions:

We have now added a paragraph and a new figure to our manuscript (subsection “Estimation methods can be used for species classification of acoustic recordings”, second paragraph, Figure 3) that discusses the differentiability of wingbeat frequency distributions in relation to their overlaps between species. We employ both the Jensen-Shannon Divergence (JSD) metric, which is related to the mutual information between statistical distributions, and our classification algorithm applied to subsets of the data used to create these distributions, to quantify the degree to which distributions can be distinguished from each other.

Counts needed to generate a distribution:

We have added a paragraph and a new figure to our manuscript (subsection “Mosquito species have characteristic wingbeat frequency distributions that are measurable using mobile phone recordings”, second paragraph, Figure 3—figure supplement 1) discussing the minimum number of counts needed to ensure that a species frequency distribution is well sampled, and verify that this is indeed the case for the data we present in Figure 3. We do this by showing statistical convergence between the species distributions and random subsets of increasing lengths drawn from them, and demonstrating that convergence (measured by complete overlap between point mass functions, and JSD showing identical information content) occurs at a small fraction (below 0.2 in most cases, below 0.5 in all cases) of the total number of samples in each species distribution.

Bandwidth required for a single flight trace:

We recommend a recording duration of at least 1 second of sound, since we found empirically that this is the minimum duration for which a recording must be made to be well sampled (subsection “Mosquito species have characteristic wingbeat frequency distributions that are measurable using mobile phone recordings”, second paragraph). However, in practice, we obtained field recordings with varying lengths between 0.1 and 13 seconds, for which we observed no relationship between correct classification and flight trace duration.

We analyzed the relationship between the duration of a flight trace and the probability of correct classification for 82 recordings from wild *Aedes sierrensis* mosquitoes (Figure 4—figure supplement 2). When compared with the other local species in our database – *Culex pipiens, Culex quinquefasciatus, Culex tarsalis*, and *Culiseta incidens* – we observed that recordings are correctly or wrongly classified solely as a function of mean wingbeat frequency, based on the overlap of the recording with the other species. The duration of the flight trace did not affect the probability of classifying it correctly as *Aedes sierrensis* or wrongly as another species.

Thus, we find that although bandwidth is not a critical parameter for classification in our current algorithm, it is nevertheless better to record for longer, to collect more data that could potentially be used to strengthen confidence in classification using other algorithms that take this into account.

-Along the same lines, reviewers concurred that results as presented do not really demonstrate that given a single measurement the species can be differentiated. Reviewers suggested that perhaps the authors could frame the question differently, instead of: given that the mosquito belongs to species A or B, can we work out which? The problem with this is that of course there are multiple possible species a sample could belong to. Presumably a single measurement would not be able to demonstrate which species a sample came from, but if they record the mosquito for long enough, can the species be determined by comparing the distribution of measurements against their reference bank?

We have followed the reviewers’ suggestion that the problem of species identification be framed as a problem of multi-class classification. We have developed a classification algorithm based on Maximum Likelihood Estimation, to identify the most probable species that could have produced the observed frequencies in a recording, given prior knowledge of the characteristic frequency distributions for each species. We have now provided a detailed discussion of this classification approach, its validation, and application to our dataset, in subsections “Estimation methods can be used for species classification of acoustic recordings” and “Acoustic data can be combined with metadata to improve classification accuracies”, and in subsection “Testing and validation of Maximum Likelihood Estimation (MLE) algorithm” of the Materials and methods section, supported by the new Figure 3 and Figure 3—figure supplement 1.

We clarify here that our approach does not treat a given flight trace as a single observation corresponding to some averaged frequency value, but as a collection of frequency values observed in each sample window that the trace is divided into (subsection “Estimation methods can be used for species classification of acoustic recordings”, first paragraph). This allows to frame the question of how long to record for in terms of the number of samples required, which we have addressed in our first response, and in subsection “Mosquito species have characteristic wingbeat frequency distributions that are measurable using mobile phone recordings”, second paragraph and Figure 4—figure supplement 2 of our manuscript.

-Also reviewers wondered: with the wild caught Anopheles mosquitoes in Madagascar was it not possible to differentiate the species? Much is made earlier in the text of being able to differentiate members of the Anopheles gambiae complex yet this isn't done in the wild. Was this because species were not identified in the field or because they could not be distinguished? Either way this should be highlighted.

We have clarified in the text (subsection “Mobile phones can be effectively used to map mosquito activity through user-driven acoustic surveillance”, fifth paragraph) that field identification of wild caught mosquitoes in Madagascar was confined to separation of *Culex* and *Anopheles* mosquitoes at the genus level only.

Although acoustic distinctions were very clear between *Culex* and *Anopheles* mosquitoes (see aforementioned paragraph), we did not have other means to distinguish them in the field according to species at the time of our study. Because of permit-related limitations in transporting mosquitoes, most samples could not be brought back to our lab.

-Discussion of how does the inter-species variability relate to the intra-species variability. Comparison of wild caught mosquitoes shows that the inter-quartile range is substantially different (non-overlapping) and varies between different sized individuals in the same population. The magnitude of this variation seems greater than the difference between species the authors say it can differentiate though this is not mentioned. Comparison of the wild and colony mosquitoes is not done and should be discussed.

We expanded our analysis of wild caught mosquitoes to survey wingbeat frequencies for more than 80 individual recordings of *Aedes sierrensis* (Figure 4—figure supplement 2). We summarize our findings in the text (subsection “Mobile phones can be effectively used to map mosquito activity through user-driven acoustic surveillance”, second paragraph), namely that frequency spreads are much tighter for individual flight traces as compared to species level distributions, and that mean frequencies are tightly clustered for the given species. Although the inter-quartile range varies and does not overlap for a few individuals, the absolute magnitude of this difference (order of a few tens of Hz) is relatively small compared to the spread of around 100-200 Hz for a given species distribution.

However, we cannot make direct comparisons between the effects of inter-species and intra-species variations on classification, as these are independent contributions to classification errors. Differentiation or classification is an inherently probabilistic operation. Hence the strongest statement we can make is only in terms of the chance of classifying a recording from a given species correctly. This is a function of both the probability of finding that set of frequencies within that species, and of the overlap of the species’ frequency distribution with other species – the two being completely unrelated factors.

It is not necessary that distinct species should have frequency distributions very different from each other. Indeed, this is not the case (Figure 3), with some species pairs having highly similar frequency distributions. It is quite possible that the differences between some individuals of one species will be greater than or equal to the differences between that species and another.

In the cases of a few species with high overlap, correct classification of individual recordings is also very challenging, with a high probability of making mistakes. Conversely, it is also possible that there are some outliers within a species, which are consistently misclassified as another. These are extreme cases, and in such situations metadata or acoustic parameters other than wingbeat frequency need to be brought into the picture, to create a distinction between species. There is also much scope for developing Bayesian approaches to solve this problem, to continuously refine classification accuracy based on prior observations and knowledge; however that lies outside the scope of this first manuscript.

-One reviewer also noted that the statement that Aedes aegypti and Anopheles gambiae mosquitoes can be differentiated by their biting time is wrong. "To my knowledge though An. gambiae are unlikely to bite in the day both are highly active at dawn and dusk and Aedes can certainly be found biting at night. Without the use of this meta-data these mosquitoes could not be distinguished acoustically and this should be highlighted."

We note the reviewers’ concern here, and have appropriately reworded the manuscript (subsection “Acoustic data can be combined with metadata to improve classification accuracies”, last paragraph) to indicate a more nuanced approach.

Chen et al. J. Insect Behav. 2014 first introduces the concept of using time of recording to improve classification accuracy, specifically in the case of *Aedes aegypti.* Yet, we agree with the reviewers that time cannot create an absolute distinction between the two species. Still, we point out that it nevertheless presents opportunities for partially improving classification between the otherwise difficult-to-distinguish pair of *Aedes aegypti* and *Anopheles gambiae*. We expect that as we collect more field data to build a more complete and fine-grained picture of the activity of these two species over time, it may become possible to build a suitable filter to differentiate the two based on this data. At present though, we have modified our manuscript to reflect the promise/potential in using time as a distinguishing feature, without conveying that this is a hard and fast rule.

-Inter vs. Intra colony and variability: The Materials and methods shows multiple colony mosquitoes of each species though sometimes these appeared grouped together in Results (e.g. Anopheles arabiensis) though at other times they were separated (e.g. An. quadrimaculatus, Figure 3). Nowhere in the manuscript is inter-colony variation mentioned and it needs to be as it is immensely important. If new calibration datasets are needed for each mosquito sub-population the potential for citizen science is massively diminished. The inter-colony, intra-colony and inter-individual variability all needs to be discussed and ideally separated as currently they are all grouped together making it impossible for the reader to determine the accuracy of the method. Comparison of the wild and colony mosquitoes is not done and should be discussed.

Here, we clarify that only a single colony of *Anopheles quadrimaculatus* was studied, but it has been presented adjacent to the similarly named *Anopheles quadriannulatus* – also a single colony. However, in the specific cases of *Aedes aegypti, Anopheles gambiae*, and *Anopheles arabiensis*, we had recorded multiple colonies, and Figure 3 represents the aggregate distribution from all colonies grouped together.

Based on the reviewers’ feedback, we have incorporated a new analysis of inter-colony variation in the manuscript for these three species (subsection “Mosquito species have characteristic wingbeat frequency distributions that are measurable using mobile phone recordings”, last paragraph, new Figure 3—figure supplement 1). Here, we discuss the similarity between colonies for *An. gambiae* and *Ae. aegypti*. However, we also note that frequency distributions vary significantly for *An. arabiensis* colonies. While two of the colonies presented for *Ae. aegypti* are lab-reared, one is reared from eggs collected in the field, allowing for comparison between populations of the same species that have been completely reproductively isolated for at least over a hundred generations (age of the lab-reared colonies). We discuss the implications of this, indicating that while some species may be classified using universal frequency databases, others may need local distinctions to be made.

We have also added a discussion on inter-individual variability between wild mosquitoes, using around 80 individual recordings of wild *Aedes sierrensis* mosquitoes collected by users at a field site (subsection “Mobile phones can be effectively used to map mosquito activity through user-driven acoustic surveillance”, second paragraph, new Figure 4—figure supplement 2). We show that most flight traces have mean frequencies clustering within a relatively narrow range, and spreads (5^th^ to 95^th^ percentile) of below 10% of the mean frequency (Figure 4—figure supplement 2), demonstrating reasonable levels of similarity between individuals in the wild.

We disagree that the potential for citizen science is largely diminished if multiple calibration datasets are required. We point out that acoustic records do not even exist for most species, and out work presents the first systematic groundwork done to survey a greater number of species across different areas. Our proposed tool is accessible to the general public, and calls for a large number of users, making local distributions possible. Acoustic wingbeat data also provides a quantitative framework for standardized data collection across many studies, because of which any number of decentralized local surveys can be stitched together to create an increasingly detailed picture of mosquito populations and their associated sounds in different locations. More people engaging in data collection also improves the overall performance of the system. For example, local vector control districts or field entomologists at any location can create annotated databases relevant to their location, using the same framework and approach. With an increasing number of such local databases, it may become possible to study these variations within species on a much larger scale than hitherto possible. We lay the basis for such an effort in our current manuscript, with the outlook that much larger datasets will be collected via community efforts in the future.

B) Classification details and methods: the reviewers suggest the authors consider advanced methods to improve classification (e.g. machine learning) and provide more details on frequency distributions and measurement requirements to around the classification and classification accuracy. Details:-Classification accuracy (albeit unblended) could and should be presented more quantitatively for the data presented in Figure 3 – at least at the pairwise level (proportion of individuals correctly classified in a mix of 2 species). A more sophisticated analysis might be to sample potentially collocated species from the data in 3A and present multi (>2) species classification accuracy.

We recognize the importance of presenting quantitative classification approaches in our work as the primary concern of the reviewers. We have now implemented this suggestion with a new detailed treatment, in subsections “Estimation methods can be used for species classification of acoustic recordings” and “Acoustic data can be combined with metadata to improve classification accuracies”, new Figure 3, and new Figure 3—figure supplement 1.

Our approach has been briefly described in our second response to point A. In brief, we have chosen a Maximum Likelihood Estimation approach to classify recordings from female mosquitoes of the 20 different species we survey in our work. Here, we note that prior work that describes classification algorithms typically focus on differentiating a smaller number of classes (between 2 to 8). This even includes males of the same species considered as distinct classes, which have inherently divergent frequencies as compared to females and are highly unlikely to be misclassified as such. Thus, our current algorithm demonstrates classification among a greater number of species than has been achieved so far. We detail the performance of this algorithm on various metrics, including performance on bootstrapped validation data, performance on new test data, improvement of classification with the inclusion of location metadata, and dependence of classification accuracy on the duration of the recording.

-Machine learning methods should be considered to optimise classification accuracy.

In our current work, we have used a Maximum Likelihood Estimation (MLE) approach to classification, as discussed in our second response to point A and our first response to point B. We used this approach as it is easy to implement and shows good classification accuracies, while being highly appropriate for the nature of the problem we are solving.

There is some existing literature that focuses on developing algorithms for the automated identification of insect species using wingbeat frequencies (Moore et al. J. Econ. Entom. 1986; Moore J. Insect Behav. 1991; Chen et al. J. Insect Behav. 2014). The scope of this current manuscript is to establish a new method to bring data-collection out of the lab and into hands of common users in the field. In continuing work ongoing in this area, we have already begun pursuing different machine learning approaches to acoustic species classification, based on identifying new features that characterize mosquito sounds, or developing new techniques to classify spectrograms. The MLE algorithm presented here sets a benchmark for future algorithms to match or outperform.

-One reviewer also noted that it was not clear why the higher harmonics/overtones (Figure 1, Figure 2) were being thrown away in the analysis presented. Even simple random forest methods trained on relative amplitude/frequency pairs (of course other features could be selected, such as measures of spread across a trace) for the dominant frequency and harmonics might offer considerably greater classification accuracy than just the mean dominant frequency.

We would like to clarify that our initial analysis did not explicitly mean to discard higher harmonics or disregard other acoustic properties of the mosquito signal, such as number of harmonics, degree of harmonicity, spread across a trace, etc. As we had initially not explicitly discussed classification algorithms, we instead highlighted fundamental frequency as the most common identifier used in the literature to classify mosquitoes, widely presented in other works dealing with insect identification (Reed et al. Genetics 1942; Sotavalta Acta Entomol. 1947; Sawedal and Hall Entomol. Scand. Suppl. 1979; Schaefer and Bent Bull. Entomol. Res. 1984; Unwin and Ellington J. Exp. Biol. 1979; Moore et al. J. Econ. Entom. 1986; Moore J. Insect Behav. 1991; Chen et al. J. Insect Behav. 2014).

We think that the reviewers’ suggestion of selecting other characteristic features that can identify mosquito sound is an important consideration that merits further investigation. We are currently working on new algorithms and analysis of user generated mosquito acoustic data in this regard, to identify such features using data mining methods and incorporate them into future classification algorithms, particularly as we explore more and more mosquito species. Collecting field data from a wide group of people and instruments is the key to identify robust predictors as we expand our test data sets.

However, we would like to point out here that we have also chosen to specifically exclude the relative amplitude for harmonics at different frequencies as a potential identifier. Three factors related to the recording techniques we use – mobile phone microphone frequency response, possible confinement in cups or bottles, and field recording against variable background noise – can sometimes alter the ratio of amplitudes for the first two harmonics. This is particularly evident in these cases:

1) The mobile phone microphones used to collect the acoustic data have frequency response curves that can vary across phone models. For mosquitoes with low frequency wingbeats, some phones like the Xperia Z3 Compact tend to amplify the first overtone as opposed to the fundamental frequency – an effect that can be observed in Figure 1—figure supplement 2.

2) In cases where mosquitoes are confined in a bottle or cup, if the frequency of the mosquito’s first overtone is closer to the resonant frequency of the cup, the first overtone gets amplified more than the fundamental frequency. We have provided a spectrogram giving an example of this in Author response image 1, for two *Aedes sierrensis* mosquitoes recorded in the field while confined in 50mL Falcon tubes.

3) In cases where low frequency ambient noise dominates and the mosquito’s lower overtones are drowned out by the noise, it is sometimes easier to identify higher overtones as their signal-to-noise ratio is improved by the lack of noise at those frequencies. Author response image 2 shows a situation where the second overtone is most easily distinguishable, and our processing algorithm tries to extrapolate fundamental frequency based on detecting the second overtone.

**Author response image 2. respfig2:** 

Since our method should be able to deal with data from a variety of species, collected on many different phone models having slightly varying frequency response curves, in highly variable field environments, we cannot consider a feature that is selectively distorted in some potentially unknown subset of the data.

C) Experiments: The reviewers suggested some experiments to demonstrate that it is possible to discern relative abundance would be helpful. While time may not permit a systematic field study, we offer the following strongly suggested approach for a blinded test (please note it is not the reviewer's intent for these experiment(s) to take longer than the revision time, if the authors do not think these are feasible in the allotted time period please justify why):-Replicate experiments where the authors mix (at known proportions) individuals from several vector species in an experimental large cage setting, and then get researched blinded to the species mix to (a) sample wing beat frequencies from multiple (or all) insects, and (b) analyse the results to determined predictive/measurement accuracy. If mixing mosquitoes is too hard, then sequential recordings of a mix of mosquitoes in the same environment and blinded analysis might suffice.

We have followed the reviewers’ suggestion to include a blinded analysis of mosquito recordings randomly chosen from among various species, all collected in the same environment. This test dataset is a collection of recordings from seventeen species in our database, acquired using different phones other than the one used to create their species frequency distributions (subsections “Estimation methods can be used for species classification of acoustic recordings” and “Acoustic data can be combined with metadata to improve classification accuracies”, Figure 3, new Figure 3—figure supplement 1).

We also present analyses of our classification accuracy with and without a location filter applied to this data. Although classification is not perfect, having an overall accuracy of 35% without (Figure 3—figure supplement 1) and 65% with a location filter (Figure 3), these accuracies are comparable to the inherent limits of classifiability for each species obtained from an analysis of data bootstrapped from the species frequency distributions (subsection “Acoustic data can be combined with metadata to improve classification accuracies”, first paragraph, Figure 3).

D) Range:-The reviewers noted that given the limited range as described (5cm), it would be helpful to demonstrate that the approach works (or sensitivity of outcomes) if the orientation or distance between the microphone and mosquito are not ideal.

The limitations of mobile phone hardware enforce requirements on a minimum range and specific orientation when recording mosquitoes. We have reworded our manuscript to explicitly state the importance of staying within the range of sensitivity for sufficient signal-to-noise ratio, and provide additional guidelines to the user for maintaining proper orientation and distance while recording (subsection “Mobile phone microphones are comparable to studio microphones in recording mosquito wingbeats at up to 100 mm”). We include supplementary movies that explicitly provide a visual demonstration of recording for readers.

Mobile phone microphones are inherently short-range sensors, as they are typically optimized to pick up sound from a person speaking close to the mouthpiece. This also implies that mobile phones will only be able to pick up sounds from mosquitoes within this range, and cannot passively acquire signals from far away mosquitoes without being overwhelmed by noise. For this reason, our recording methods require active human involvement in bringing the phone sufficiently close to the mosquito. At the same time, this is an advantage because users are directly engaging/making decisions on what to record from; removing other insects that are often found in autonomous systems.

For the effects of orientation, we consider two conditions – the orientation of the insect in flight with respect to the microphone (which is not under the control of the user), and the orientation of the phone that directs the mosquito towards or away from the insect (which is completely under the control of the user).

For the first aspect, we refer to work done by Arthur, Emr, Wittenbach and Hoy (J. Acoust. Soc. Am. 2014) which studies the effect of orientation on amplitude and phase of harmonics for tethered mosquitoes. The amplitude of the wingbeat sound has a directional variation, being about 10dB louder ahead of and behind the mosquito. However, this is not a significant factor affecting our data, since we make recordings in free flight where both the orientation and distance of the insect with respect to the mobile phone is highly variable during flight sequences. For the typical amplitudes of 40-60dB that we record for mosquito sound (as measured in the lab on free-flying caged mosquitoes), 10dB does not reduce the sound below detectable levels.

For the second aspect, the orientation of the phone itself is critical to improving the signal-to-noise ratio of the recording, much like the range. It is important that the user orient the primary microphone towards the mosquito as directly as possible, as the sounds that are not in the line of sight of the primary mic are generally considered noise by the phone (dual microphone noise reduction technology). The general location of these microphones is commonly known to users, and we also describe how to look for them in our instruction sheets.

Thus, we clarify that this approach will not work under non-optimal range and orientation of the phone, and that users must take care to actively maintain favourable conditions for recording. However, we have typically found this to be quite easy to do, even for free-flying mosquitoes in the field, and have received similar feedback from our volunteers (picked from the general public). Hence, we do not consider this limited range an impediment to acoustic data collection, but merely a caveat to bear in mind to obtain high quality data. Users also get feedback by listening back to what they recorded; allowing them to learn from the process.

E) Data and accessibility:-The authors are encouraged to follow open data norms and provide the raw data that went into the figures in the main manuscript as supplements to the paper.

We have followed this recommendation and posted all our data corresponding to the 20 species in our database. This includes recordings made with several different phone models, and for multiple colonies. We have posted this data both on the Dryad data repository as recommended (http://dx.doi.org/10.5061/dryad.98d7s), as well as on our dedicated website for the project (abuzz.stanford.edu). We have also included information on how to access this data, in the supplement to the paper (subsection “Downloadable data package”).

-As well, in line with the aims of the paper the reviewers agreed that it would be great if the authors can elaborate on how Joe Public can actually get started with this (release data and app).

As per the recommendation of the reviewers, we have included a graphical supplement to the paper (Figure 1—figure supplement 2), which provides specific guidelines to users on how to record and upload acoustic data to our servers. Further, we have provided a link in our manuscript to our dedicated website for the project (subsection “Downloadable data package”), where are setting up a portal for users to upload data, receive feedback and view maps of mosquito uploads. We have also included new videos that demonstrate how to record mosquitoes.

As this is an ongoing project, we are continuing our efforts to set up a platform for user engagement through the website and an app. Both are currently under construction at the moment, in anticipation of scale-up and a public release in the near future. Using the web interface, all users can already start uploading data to our platform.

F) More discussion or real applications in terms of mosquito species and limitations:-The reviewers note that this approach will work well for anthropophilic mosquitoes, but not as well as animal feeders esp. avian feeders that are not attracted to man.

We acknowledge the reviewers’ observation that our approach to surveillance is biased towards anthropophilic mosquito species, as it requires human intervention in recording mosquito data. We have also specified this in our manuscript (Discussion, third paragraph).

However, we note that most methods of surveillance in use today, such as light traps or ovitraps, are usually biased towards certain species of interest. The data collected by our phone-based approach can be complementary to data that these other methods yield, stitching together a broad view of mosquito activity with a fine focus on their human interactions. We believe this can be highly effective in narrowing the focus onto potential vectors of human disease, which are typically the species of maximum interest and impact, particularly in the context of public health interventions in resource-limited areas.

Finally, automated permanent installations of cellphones recording mosquitoes in remote areas is not inconceivable, for approaches that need to track non-anthropophilic mosquitoes.

-As well there is high applicability to ID exotics; the authors are encouraged to augment discussion of the approach's relevance to Aedes and, for example current citizen science program in Europe doing also surveys, or towards surveillance of albos in Australia (PLoS neglected tropical diseases, 11(2), p.e0005286)-These include hard to catch and measure mosquitoes at night (when they are most active).

We thank the reviewer for pointing this out, and have included this point in our manuscript (Discussion, second paragraph). As per the review comment, we have also included references to citizen science programs with similar aims in various countries.

Will size impact the WBF? Most wild mosquitoes are smaller than lab reared. Will this be a confounder to ID? And what about temperature and age?

We have added a detailed discussion of these factors in our manuscript (subsection “Mobile phones can be effectively used to map mosquito activity through user-driven acoustic surveillance”, second paragraph, Figure 3, Figure 4—figure supplement 2), showing that size and age are not expected to affect wingbeat frequency.

Here, we also refer to recent work by Villarreal, Winokur and Harrington (J. Med. Entom. 2017), which studies the impact of both temperature and body size as independent variables affecting the wingbeat frequency of female *Aedes aegypti* mosquitoes, under controlled lab conditions. To summarize the results from this work, temperature is directly and linearly correlated with wingbeat frequency increasing by around 8-13 Hz/°C, whereas body or wing size does not impact the wingbeat frequency. To control for the effects of temperature on our data, we intend to collect data about local weather from users (which is also often known given the location data). Here, it is also possible to follow suggestions by other studies that automate the recording of insects in field conditions (for e.g., Chen et al. J. Insect Behaviour 2014), where linear models can be used to map frequencies recorded under ambient conditions to a standardized temperature prior to processing and identification.

The wingbeat frequency of a mosquito has also been found to remain constant with age 2-3 days post eclosion (PhD thesis by R. Costello, Simon Fraser University 1979). Since mosquitoes are actively host-seeking and most likely to encounter humans after around 3 days of age, we expect not to see significant effects of age on wingbeat frequency in our data from human-mosquito encounters. Hence, we have not explicitly controlled for this.

-The reviewers noted that this would particularly be an appropriate tool to survey for exotic species, especially where the species of concern is known (e.g., Ae. albopictus). It would also be a great tool to measure populations for rear and release programs such as wolbachia and sterile males. In this instance, we need to know the wild population so that effective numbers can be released. Phones might be a great way to do this.

We thank the reviewer for pointing this out, and have mentioned this point in our manuscript (Discussion, third paragraph). We also believe in building open ended tools, with the hope that both the scientific community and general public can use these tools in context-dependent programs around the world.

-Oc. sierrensis should be Aedes sierrensis.

We have changed this as necessary in the manuscript.

-Also, how did they measure WBF? In a bag, or free flying?

We have added mentions in the text to explicitly specify this. In most of the lab and field collected data, wingbeat frequencies are measured in free flight. For the lab data presented in Figure 3, the mosquitoes are recorded in free flight within a cage. For the field data presented in Figure 4, the mosquitoes were in free flight in the wild around the human host performing the recording. In a very few cases, such as the data for *Culiseta incidens* presented in Figure 3, the mosquito was trapped in a bag prior to recording.

We also however clarify that we recommend confining mosquitoes in bottles or cups rather than bags (subsection “Mobile phones can be effectively used to map mosquito activity through user-driven acoustic surveillance”, first paragraph, Figure 1—figure supplement 2, Video 1–Video 3). This is based on our more recent experiences and feedback from users, as the accidental crumpling of bags during recording introduces noise that often severely obscures the mosquito signal. Also, cups act as simple resonators amplifying the signal and do not introduce any effects on mosquito flight because of static charge on the surface (unlike plastic bags, which sometimes have this tendency). In the future, we will quantify these effects.

-In several places you mention species-specific wingbeat frequencies. They are not really species specific as there is some overlap. Please remove species-specific.

Our intent in using the term “species-specific” wingbeat frequencies was to convey that the wingbeat frequencies for a given species characteristically fall within a specific range. We have adopted this terminology from previous investigations of acoustic mosquito classification (Chen et al. 2014). However, we note the reviewers’ point that this term can be mis-interpreted as the wingbeat frequencies characteristic of a given species being unique to that species alone, which need not always be the case. If required, we can modify this to the term “characteristic wingbeat frequencies”.